



# Short Large-Amplitude Magnetic Structures (SLAMS) at Mercury observed by MESSENGER

Tomas Karlsson[1], Ferdinand Plaschke[2], Austin N. Glass[3], and Jim M. Raines[3]

[1]Division of Space and Plasma Physics, School of Electric Engineering and Computer Science, KTH Royal Institute of Technology, Stockholm, Sweden
[2]Institut für Geophysik und Extraterrestrische Physik, Technische Universität Braunschweig, Braunschweig, Germany
[3]Dept. of Climate and Space Sciences and Engineering, University of Michigan, Ann Arbor, MI., USA

**Correspondence:** Tomas Karlsson (tomask@kth.se)

**Abstract.** We present the first observations of Short Large-Amplitude Magnetic Structures (SLAMS) at Mercury. We have investigated approximately four years of MESSENGER data to identify SLAMS in the Mercury foreshock. Defining SLAMS as magnetic field compressional structures, with an increase in magnetic field strength of at least twice the background magnetic field strength, when MESSENGER is located in the solar wind, we find 435 SLAMS. The SLAMS are found either in regions
of a general ultra-low frequency (ULF) wave field, at the boundary of such a ULF wave field, or in a few cases isolated from the wave field. We present statistics on several properties of the SLAMS, such as temporal scale size, amplitude, and the presence of whistler-like wave emissions. We find that SLAMS are mostly found during periods of low interplanetary magnetic field strength, indicating that they are more common for higher solar wind Alfvénic Mach number ($M_A$). We use the Tao solar wind model to estimate solar wind parameters to verify that $M_A$ is indeed larger during SLAMS observations than otherwise.
Finally, we also investigate how SLAMS observations are related to foreshock geometry.

## 1  Introduction

As the supermagnetosonic solar wind encounters the planets in the solar system, a bow shock is formed at which the solar wind is slowed down to submagnetosonic velocity (e.g. Balogh and Treumann, 2013). The region upstream of such a shock which is magnetically connected to the shock is called the foreshock (Eastwood et al., 2005b). Depending on the angle between the
connecting field line and the bow shock normal ($\theta_{Bn}$), the foreshock connects to either the quasi-perpendicular bow shock ($\theta_{Bn} > 45°$) or to the quasi-parallel shock ($\theta_{Bn} < 45°$). At supercritical shocks, a portion of the solar wind plasma is reflected, and travels upstream into the foreshock region. At Earth, an important effect of this is that the reflected ions together with the original solar wind ions trigger the ion-ion right-hand resonant beam instability, resulting in the excitation of foreshock ultra-low frequency (ULF) waves (Gary, 1991). At Earth, these fast magnetosonic waves typically have a period of around 30
s, and are typically observed in the quasi-parallel part of the foreshock (e.g. Hoppe and Russell, 1983; Eastwood et al., 2005a; Burgess et al., 2012). While these 30 s waves are the most studied ones, other types of waves with other periods (10 s, 3 s, and 1 s) are also observed in the Earth's foreshock (e.g. Eastwood et al., 2003; Blanco-Cano et al., 1999, 2011).





In the terrestrial foreshock, the further interaction of the 30 s waves with the reflected ions, and the modified particle distributions resulting from the ion-ULF wave interaction are believed to result in non-linear growth of the waves. The ULF waves may the develop into isolated spikes in the magnetic field amplitude, reaching amplitudes of several times that of the background magnetic field. Such monolithic structures are commonly known as Short Large-Amplitude Magnetic Structures (SLAMS) (Schwartz and Burgess, 1991).

SLAMS propagate in the sunward direction in the solar wind frame, but since their phase velocity is smaller than the solar wind velocity, the net result is that the are convected downstream, towards the bow shock (Schwartz et al., 1992). While the SLAMS are convected downstream, their amplitude increases. Mann et al. (1994) showed that possibly (the study was based on very few events), the phase velocity of SLAMS increases with amplitude, leading to the idea that large amplitude SLAMS may attain a phase velocity that has the same magnitude as the oppositely directed solar wind velocity. This would result in large-amplitude SLAMS standing in the bow shock frame of reference, making the SLAMS building blocks of the quasi-parallel bow shock (Schwartz et al., 1992). The properties of SLAMS may therefore be critically important to understand the nature of quasi-parallel shocks. This includes local shock reformation (e.g. Johlander et al., 2022, and references therein), as well as the formation of magnetosheath jets, either via the formation of bow shock ripples (Plaschke et al., 2018), or as a direct consequence of shock reformation (Raptis et al., 2022).

In spite of the possible important roles that SLAMS may play, many of their properties as well as their generation mechanisms are quite poorly understood, even at Earth. At other planets the knowledge about SLAMS is very rudimentary. Collinson et al. (2012) have presented three isolated compressive magnetic field structures in the Venus foreshock, with properties similar to terrestrial SLAMS, and Bebesi et al. (2019) presented four SLAMS observations from the upstream region of the Saturn quasi-parallel bow shock, demonstrating that SLAMS can exist in connection with both intrinsic and induced magnetospheres. Unambiguous observations of SLAMS have not been reported from any other planets, although the solitary magnetic structures in the foreshock of Mars reported on by Chen et al. (2022) share many properties of terrestrial SLAMS, and may in fact be such.

The question of the possible existence of SLAMS in the Mercury foreshock has been raised before. Sundberg et al. (2013) studied a passage of the MESSENGER spacecraft through the magnetosheath and quasi-parallel foreshock region. They observed what they called 'large-amplitude magnetic structures', although they did not give any number characterizing the amplitudes. They did not interpret the structures as SLAMS, rather as a larger-scale cyclic reformation of the bow shock. They argued that due to their clear cyclic behaviour, with similar amplitudes for each consecutive magnetic field maximum, was not consistent with the often observed monolithic structures of SLAMS (Schwartz et al., 1992). They also argued that since the same clear cyclic behaviour was observed in the downstream magnetosheath, and even magnetosphere, the structures had to have large cross-section, since a patchwork of smaller-scale, SLAMS-like structures would not produce such a stable periodicity in the downstream waves. Furthermore, the polarization of the structures were the same as the upstream ULF waves, which is not typical for SLAMS at Earth. Also Karlsson et al. (2016) argued that the rarity of magnetic compressive structures ('paramagnetic plasmoids' (Karlsson et al., 2015)) in the magnetosheath, which at Earth are interpreted as remnants of SLAMS that have crossed the bow shock, made it unlikely that SLAMS existed at Mercury.



The purpose of this paper is to revisit the MESSENGER magnetometer data set (Anderson et al., 2007), to make a comprehensive search for SLAMS at Mercury. At this point it is important to have a clear definition of what constitutes a SLAMS. Schwartz et al. (1992) defines SLAMS as structures having 1) short duration, 2) large amplitude with $|\mathbf{B}(t)|$ a factor of at least 2, and typically 3 or more, above the background field, 3) well-defined single magnetic structure, rather monolithic in appearance. Lucek et al. (2002) identify SLAMS as structures having a magnetic field magnitude of at least a factor 2.5 of the background magnetic field, with durations of the order of 10 s or. Plaschke et al. (2018) uses a classification based on the amplitude of the foreshock structures, with ULF waves having amplitudes of $\Delta B/B_0 \sim 1$, somewhat steepened waves, 'shocklets' having $\Delta B/B_0 > 1$, and SLAMS having $\Delta B/B_0 > 2$. Here $B_0$ is the background magnetic field strength, and $\Delta B$ is the deviation from that background. The same definitions for shocklets and SLAMS are used by Wilson III et al. (2013).

In this study we will define SLAMS based exclusively on the magnetic field amplitude, requiring that $\Delta B/B_0 > 2$. Properties like duration and appearance will rather be considered as a result of the investigation, and will be compared to other SLAMS observations. The details of calculating $\Delta B$ and $B_0$ will be given below.

## 2 Data and method

### 2.1 MESSENGER data

For this study, we mainly use magnetic field data from the magnetometer (MAG) onboard the NASA MESSENGER spacecraft (Anderson et al., 2007). For consistency, we only use data sampled at 20 samples per second. Data from two weeks after orbit insertion to the end of mission are used (2011-04-01–2015-04-29). Magnetic field data is presented in the MSO coordinate system (Mercury Solar Orbital), where $x$ points towards the sun, $z$ points northward, perpendicular to the planetary orbital plane, and $y$ completes a right-handed, orthogonal system.

Ion data from the Fast Imaging Plasma Spectrometer (FIPS) (Andrews et al., 2007; Raines et al., 2011) are used for context below. However, the FIPS data are not used for detailed scientific analysis, due to the inability of the instrument to directly measure the core of the solar wind beam, because of limitations in the field of view (Gershman et al., 2012). To estimate solar wind parameters, we instead use output from the Tao solar wind model (Tao et al., 2005).

### 2.2 SLAMS definition and search criteria

In order to identify SLAMS in the MAG data, we proceed in the following way, similarly to the method used by Karlsson et al. (2021a) to identify magnetic holes in the solar wind. We first identify time intervals where MESSENGER is located in the solar wind according to the identification of bow shock crossings made by the MESSENGER MAG team (Philpott et al., 2020). For all such intervals, we define a background magnetic field strength $B_0$ by averaging the time series of the magnitude of the magnetic field, $|\mathbf{B}(t)|$ with a sliding window with a width of 180 s

$$B_0(t) = \langle |\mathbf{B}(t)| \rangle_{180\,\mathrm{s}}, \tag{1}$$





where the angle brackets indicate the time averaging. The relative deviation $\frac{\Delta B}{B_0}(t)$ from this background level is then calculated by subtracting $B_0$ from the original time series of $|\mathbf{B}(t)|$, and dividing by the background level. We also smooth the resulting signal by a 0.35 s sliding window (7 data points), to remove the highest frequency variations:

$$\frac{\Delta B}{B_0}(t) = \left\langle \frac{|\mathbf{B}(t)| - B_0(t)}{B_0(t)} \right\rangle_{0.35\,\text{s}}. \tag{2}$$

The longer window time was chosen since it averages over many waves periods of the foreshock ULF waves that may develop into SLAMS, which have been reported to be around 1-20 s (Le et al., 2013; Romanelli et al., 2020). The shorter window size was chosen somewhat arbitrarily after visual inspection showed this to not affect the main impression of the resulting SLAMS candidates, while letting the wave forms of the structures stand out more clearly.

We then identify a first set of SLAMS candidates by recording time intervals where $\frac{\Delta B}{B_0}(t) > 2$. We thereafter remove candidates where the maximum magnetic field strength during the interval was greater than 100 nT, since these are almost exclusively due to telemetry errors or other non plasma physical processes, which were clearly identifiable by visual inspection. We also remove candidates where the time resolution $\Delta t$ is not equal to 0.05 s. After this post-processing step we were left with 429 SLAMS candidates. All candidates were visually inspected to identify further possible candidates related to artefacts not related to plasma physics. No further SLAMS candidates were eliminated in this process.

### 2.3 Tao model

In lieu of in situ solar wind plasma data, as mentioned above, we have used model data to investigate the dependence of SLAMS occurrence and properties on solar wind properties. The Tao model (Tao et al., 2005) is a 1-D MHD model that uses observations close to Earth orbit, and propagates them backward or forward in time (corresponding to heliocentric distances less than, or greater than 1 AU, respectively.) We have used the values of plasma density and bulk velocity, propagated to Mercury orbit, available at the AMDA online database (Génot et al., 2021).

## 3 Results

We begin by showing a typical example of a SLAMS observation, followed by an investigation of the statistical properties of the identified SLAMS.

### 3.1 Examples of SLAMS

Figure 1 shows one hour of observations from the magnetosheath and solar wind from 2013–03-10. The bow shock (identified by Philpott et al. (2020)), is indicated by the vertical orange dashed line at 10:43:53 UTC. Before that, MESSENGER is located in the magnetosheath, which is associated with a compressed and turbulent magnetic field, as well as a heated ion population, as compared to the upstream solar wind. Upstream of the bow shock, two regions of ULF waves with periods between approximately 6 and 7 s can be seen (10:44-10:47:30, and 10:49-10:52 UTC). Collocated with the second of these





ULF wave intervals are some weak fluxes of suprathermal ions. Together these signatures indicate that the spacecraft is located in the ion foreshock. The ULF waves have a $\frac{\Delta B}{B_0} < 0.5$ and a closer inspection revealed no sign of non-linear structures, such as shocklets or SLAMS.

In contrast, in a region of a significantly depressed magnetic field magnitude (11:07-11:18 UTC), a clear excursion of $\frac{\Delta B}{B_0}$ to values of over 5 can be seen. This region is also associated with a hotter plasma than the surrounding solar wind. This might be the signature of a mildly energized diffuse population (Glass et al., 2023), although a closer investigation would be necessary to verify this. Fluctuations in this region have a $\frac{\Delta B}{B_0}$ up to about 1, and have a less clear sinusoidal character, indicating that they are evolving towards a more non-linear state. The magnetic field and particle signatures are consistent with MESSENGER

being located ion the foreshock, also for this interval.

Panels (h)-(j) show a zoomed in plot of the high $\frac{\Delta B}{B_0}$ excursion, and reveals three clearly separated peaks in magnetic field magnitude. Two of these (structures 2-3) have $\frac{\Delta B}{B_0} > 2$, fulfilling the basic SLAMS criterion, while structure 1 just falls short of this criterion, and would be classified as a shocklet in the nomenclature of e.g. Plaschke et al. (2018). The distance between the three peaks are 4.5 s and 3.3 s, respectively, broadly consistent with typical foreshock ULF periods.

For each of these peaks, we have performed a minimum variance analysis (MVA) (e.g. Sonnerup and Scheible, 1998), and have plotted a hodogram in the plane perpendicular to the minimum variance direction (which would be directed in the propagation direction of these structures, if they were plane waves.) Panels (e)-(g) show the result. It can be seen that all three structures exhibit an elliptic polarization. (Structure 1 exhibits some superposed higher frequency oscillations the middle of the structure, but this does not affect the general conclusion.) In all three panels, the minimum variance direction, which we

identify as the $k$ vector, points out of the plane of the paper. (The ambiguity of the sign of the $k$ vector has been resolved by forcing it to point in the general direction of the background field, which means that the angle $\theta_{Bk} < 90°$). This means that all three structures are right-hand polarized in the spacecraft frame, similar to observations at Earth (e.g. Schwartz et al., 1992). All in all, the three structures have similar properties to SLAMS observed in Earth's foreshock, and we conclude that these structures are likely close Mercury foreshock analogues.





**Figure 1.** Overview of observations from 2013-03-10. (a) Ion spectrogram. (b) magnetic field components in MSO coordinates. (c) magnitude of magnetic field. (d) relative change in magnetic field. (e)-(g) hodograms of the magnetic field for the three magnetic field structures shown below. B1 and B2 are the maximum and medium variation directions, the background magnetic field points out of the plane, and $\theta_{Bk}$ is the angle between the $k$ vector (minimum variance direction), and the magnetic field. (h)-(j), same as (b)-(c), but for a zoomed in time interval. Three structures are marked, corresponding to the hodograms in panels (e)-(g). The orange line corresponds to the location of the bow shock crossing given by Philpott et al. (2020)

.

## 3.2 Statistical results

The positions of the 429 identified SLAMS are shown if Figure 2, in the MSM coordinate system, and with $\rho_{MSM} = \sqrt{y_{MSM}^2 + z_{MSM}^2}$. The MSM (Mercury Solar Magnetospheric) system has has the origin offset from the planetary center by the magnetic dipole offset, but is otherwise defined in the same way as the MSO system. Also indicated in the figure are



the bow shock and magnetopause for nominal solar wind conditions (Winslow et al., 2013). The SLAMS observations are
seen to be relatively uniformly distributed along the bow shock. This is consistent with the findings of Glass et al. (2023), who
noted that foreshock ion populations were observed rather uniformly with respect to local time. This is probably due to the
more radially directed IMF at Mercury, as well as the small and dynamic magnetospheric system. The appearance of SLAMS
observations within the magnetosheath are due to the deviations in location of the bow shock compared to the typical location
shown in the figure.

In this context, we also note that the interval investigated by Sundberg et al. (2013), mentioned above, did not qualify as a
SLAMS according to our criterion, with a $\frac{\Delta B}{B_0}$ of just above 1 for the largest amplitude structures.

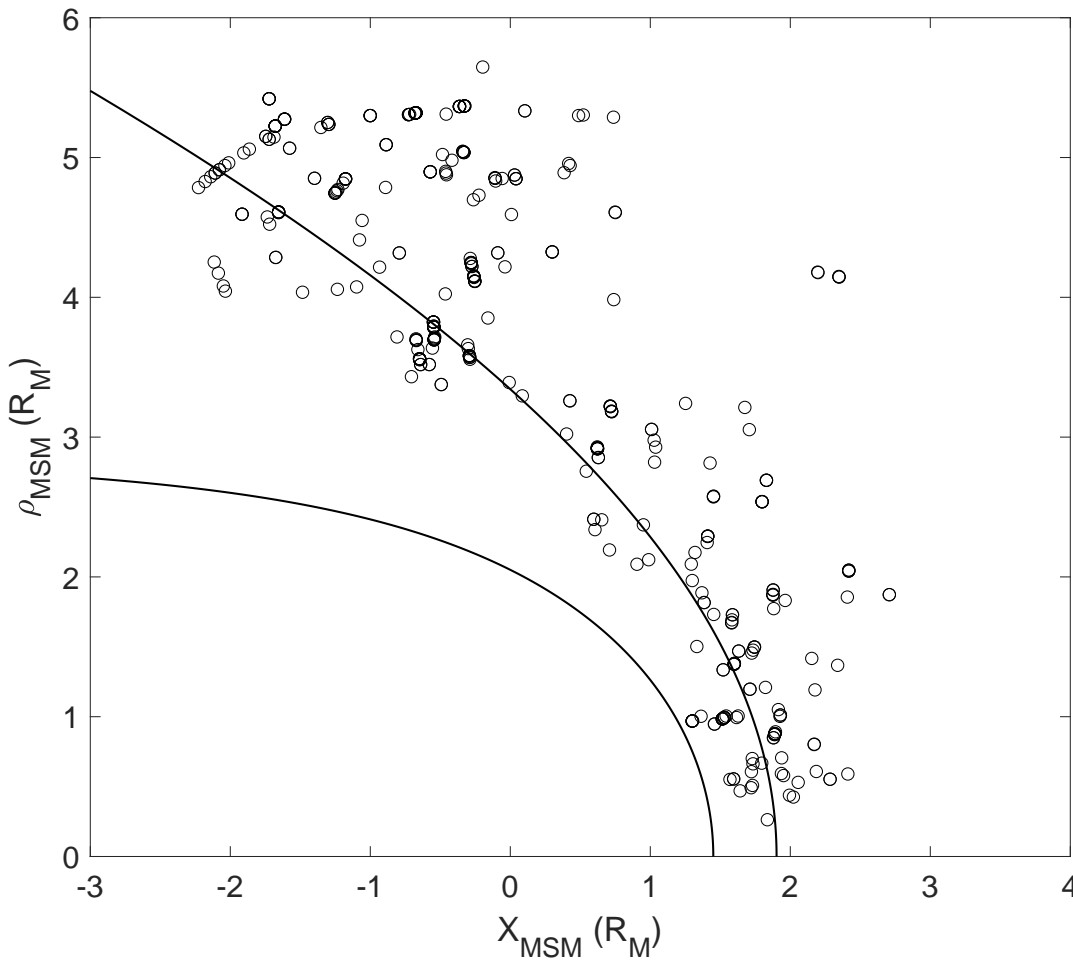

**Figure 2.** Positions of the identified SLAMS in the $x_{MSM}$-$\rho_{MSM}$ plane, in units of Mercury radii ($R_M$). Shown are also the magnetopause and bow shock for nominal solar wind conditions (Winslow et al., 2013).

### 3.2.1 Classification of SLAMS types

Before we consider some of the statistical properties of the SLAMS, we will first comment on some observations on the appearances of the SLAMS, determined by visual inspection. During this inspection, it was apparent that SLAMS appeared
in certain rather distinct forms and contexts. We have applied a classification scheme to the observations, by assigning each





SLAMS observation to one or more types. These classes are somewhat subjective, but we believe that this classification will be useful for comparing SLAMS observations from Mercury and Earth, and for future studies of Mercury SLAMS, either by the upcoming BepiColombo mission (e.g. Milillo et al., 2020), or in simulations studies.

**Figure 3.** Examples of types of SLAMS.

The SLAMS types we define are illustrated in Figure 3. The most common type of SLAMS ('Wave field') occurs imbedded in a general wavefield associated with foreshock ULF waves of periods of a few seconds up to a a few tens of seconds. A related type is the 'Boundary type', where the wave field contains compressive structures which do not $\Delta B/B_0 > 2$, except





**Table 1.** Number of SLAMS for each class.

| Class | No. of SLAMS |
| --- | --- |
| Wavefield | 351 |
| Boundary | 26 |
| Wave package | 19 |
| Isolated | 6 |
| Sharp | 17 |
| Higher frequency | 40 |

for a structure at a boundary of the wave field, outside of which the amplitude of the waves becomes very small. A further type, which we call 'Wave package', is similar to the wave field but contains only very few (2-3) wave periods. The 'Isolated' type is observed in the absence of any clear surrounding ULF wave activity. Two further types have been defined. As was discussed by Tsubouchi and Lembège (2004), at a certain stage of the growth of the SLAMS, they attain a very sharp edge at their upstream side. This is also a commonly observed property of shocklets (Hoppe et al., 1981). We have made note of a number of such SLAMS and classified them as the 'Sharp' type. Finally, large-amplitude isolated structures have also been observed in regions of wave activity of higher frequencies than those usually associated with foreshock ULF waves. We call this type 'Higher frequency'. The example shown in Figure 1 has a wave period of around 0.3 s, which can be compared to the approximate periods of 3-4 s, and 2 s for the 'Wave field', and 'Boundary' examples, respectively.

The number of SLAMS in each class is shown in Table 1. Note that one particular SLAMS can be a member of more than one class. The classification of all SLAMS can be found in the data set DOI:10.5281/zenodo.7750658.

### 3.2.2 Possible whistler precursors

As discussed by e.g. Wilson III et al. (2013), Scholer et al. (2003), and Raptis et al. (2022), SLAMS may exhibit whistler wave precursors may on their upstream side. We have observed a number of SLAMS that show a very similar behaviour, with oscillations close to the the local ion gyro frequency on the upstream edge of the SLAMS. While we do not consider SLAMS 170 with possible whistler precursors as a class of its own, we indicate 11 possible observations in the dataset DOI:10.5281/zenodo.7750658. Two examples are shown in Figure 4. The event from 2011-12-25 has a $\Delta B/B_0$ just below 2, and is another example of what may be classified as a shocklet. Note the sharp upstream part of that structures. In both cases, the possible whistler waves have a frequency (determined by visual inspection) of around twice the local proton gyro radius (evaluated with the background magnetic field, $B_0$), and have a close to circular polarization (not shown). While these possible whistler emissions need to be studied closer, they are a further indication that the SLAMS observed in the Mercury foreshock are similar to their Earth analogues.



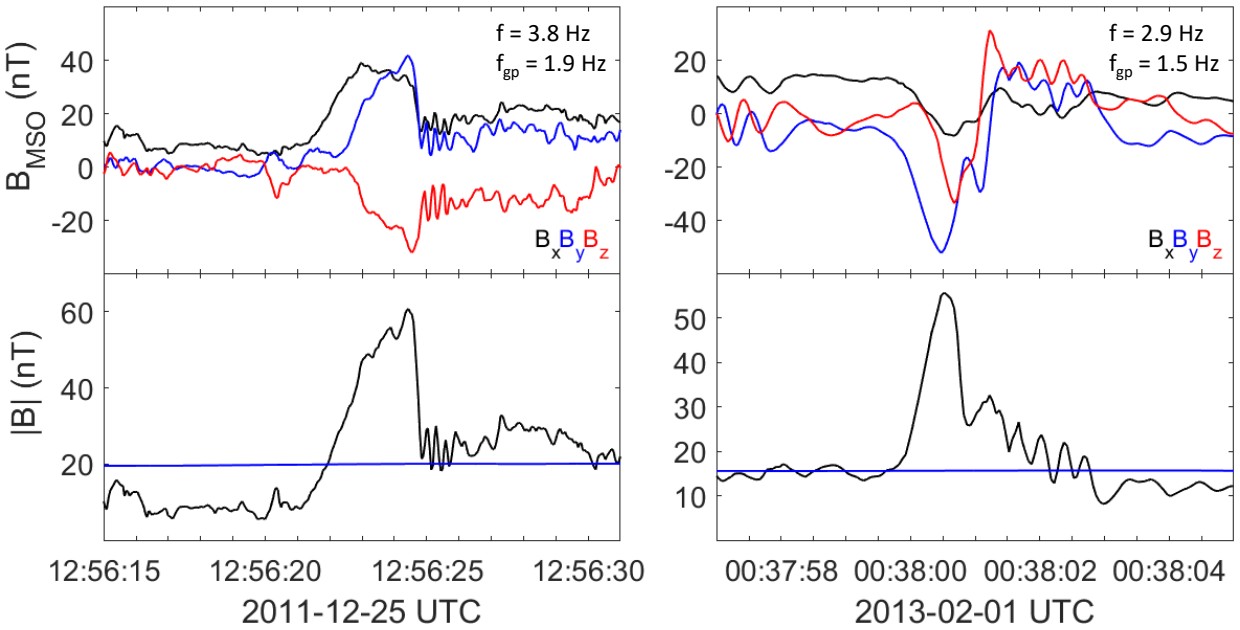

**Figure 4.** Examples of whistler-like emissions at the upstream edge of a shocklet (left), and a SLAMS (right). Also shown are the frequencies of the whistler-like waves ($f$), and the local proton gyro frequency ($f_{gp}$).

### 3.2.3 Scale sizes

Now turning to the the statistical properties of the SLAMS observed at Mercury, we first show a histogram of the temporal scale sizes in Figure 5. The scale size for each SLAMS was determined as the full width at half maximum of the difference magnetic field $\Delta B(t) = |\mathbf{B}(t)| - B_0(t)$ (without applying any smoothing). The mean and median of the distribution are 1.2 s and 0.65 s, respectively. It is difficult to convert these to spatial scale sizes, since the SLAMS are generally unlikely to convect with the solar wind speed. At Earth there is some evidence that the propagation velocity of SLAMS depend on their amplitude. This, however, was based on only 18 SLAMS observation (Mann et al., 1994), and it is anyway unlikely that the Mercury SLAMS would have the same dependence on the amplitude. A direct comparison of the temporal scale sizes may still give some information on the relative sizes of SLAMS at Earth and Mercury. No large statistics of temporal scale sizes at Earth can be found in the scientific literature, but Schwartz et al. (1992) report on durations of 5-20 s, based on a small dataset from a single day. This is also consistent with preliminary statistical investigations based on MMS (Foghammar Nömtak, 2020) and Cluster data (Mandell, 2020). We therefore tentatively conclude that SLAMS at Mercury are smaller than those at Earth, at least in the direction parallel to their propagation with respect to the spacecraft. However, uncertainties in the propagation velocities can affect this result.



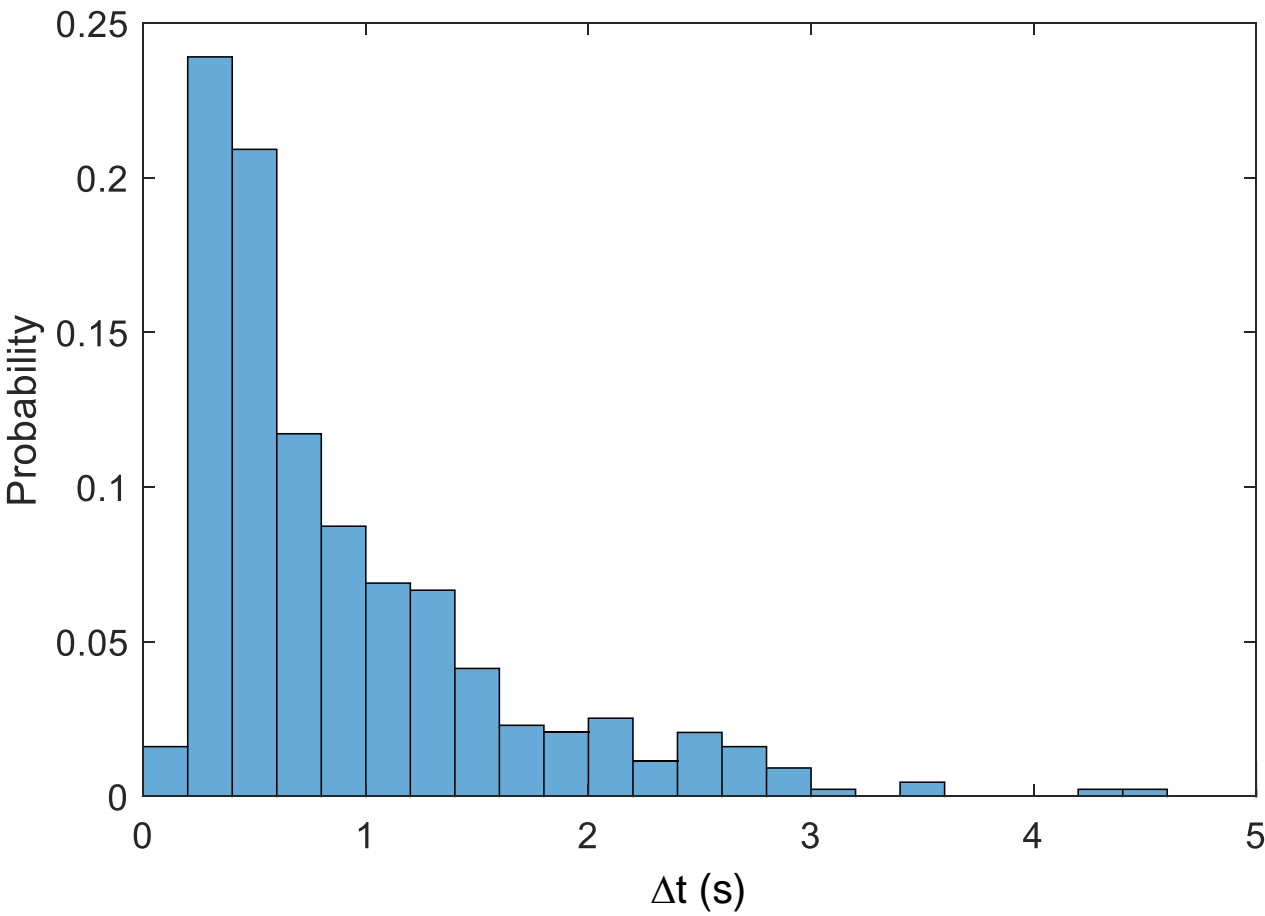

**Figure 5.** Distribution of temporal scale sizes of the observed SLAMS.

### 3.2.4 Mach number dependence

The SLAMS shown in Figure 1 were found during a time interval where the background magnetic field was significantly

smaller than for the surrounding times. On possible interpretation is that this is due to a Mach number dependence. The solar wind Alfvénic Mach number is given by

$$M_A = \frac{v_{SW}}{v_A} = \frac{v_{SW}\sqrt{\mu_0 \rho_{SW}}}{B_{SW}}, \tag{3}$$

where $v_A$, $v_{SW}$, $\rho_{SW}$, and $B_{SW}$ are the local Alfvén velocity, solar wind velocity, mass density, and magnetic field, respectively. Clearly, a lower solar wind magnetic field corresponds to a higher Mach number. Figure 6 shows the distribution of

the background magnetic field strength, $B_0$ for the SLAMS observations, compared to the distribution of the magnetic field





strength in the solar wind, determined from solar wind observations during the whole mission. It is evident that SLAMS are observed when the solar wind magnetic field is considerably weaker than the average solar wind magnetic field.

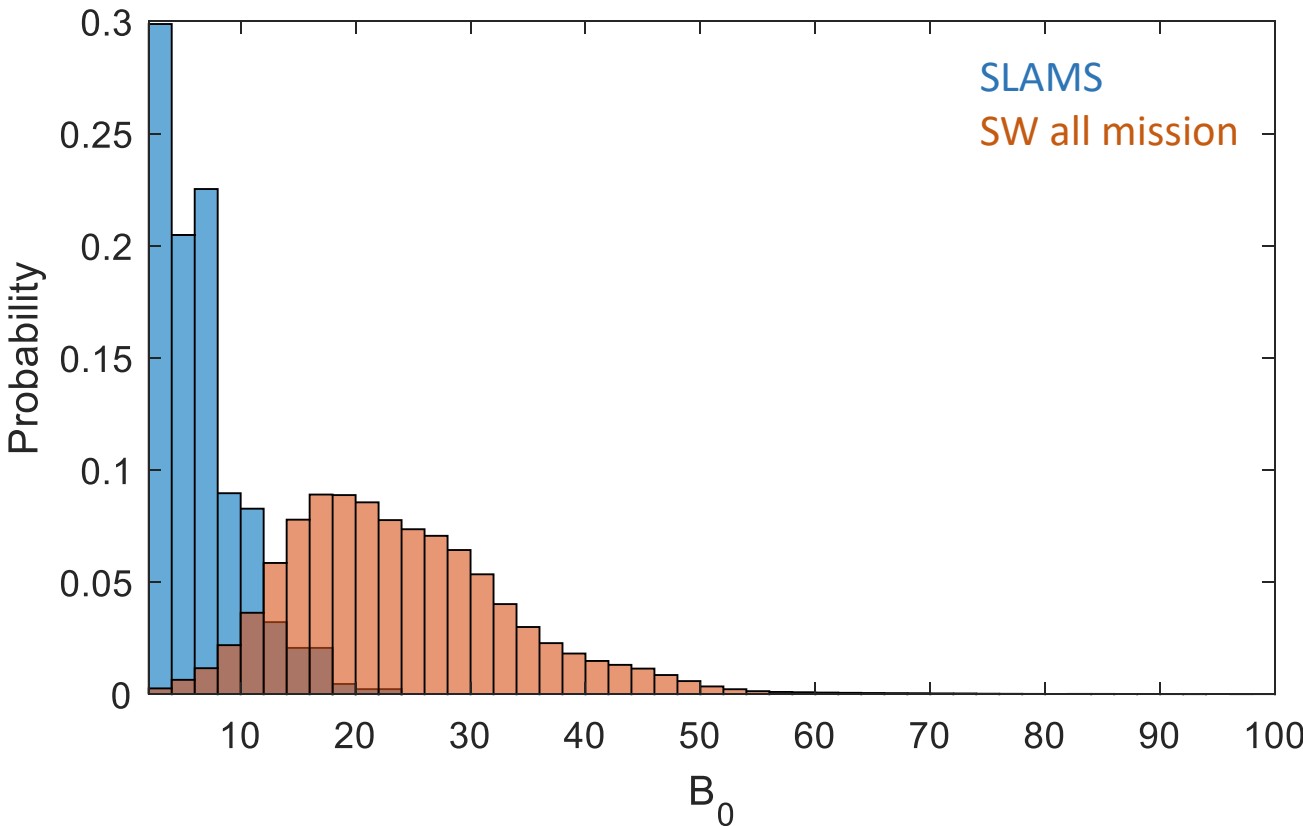

**Figure 6.** Distribution of solar wind magnetic field strength both for times when SLAMS are observed (blue), and for the whole MESSENGER mission (red).

The limitations of the FIPS instrument do not allow routine determination of the density and velocity in the solar wind, so a direct determination of the Mach number is not possible. However, we can estimate the Mach number using the Tao model.

Figure 7 shows the distribution of $M_A$ for both SLAMS and the solar wind of the whole mission. For calculating the SLAMS Mach numbers we have used the value of $B_0$ for each event, and have used the Tao model values for velocity and density closest in time to the time of the maximum $\Delta B/B_0$. The calculation in the solar wind is done in a similar way for every data point.

While the model values are associated with errors, in particular due to the limited time resolution, the results are clear and

are consistent with SLAMS observed during times of higher Mach number than usual in the solar wind at Mercury orbit.



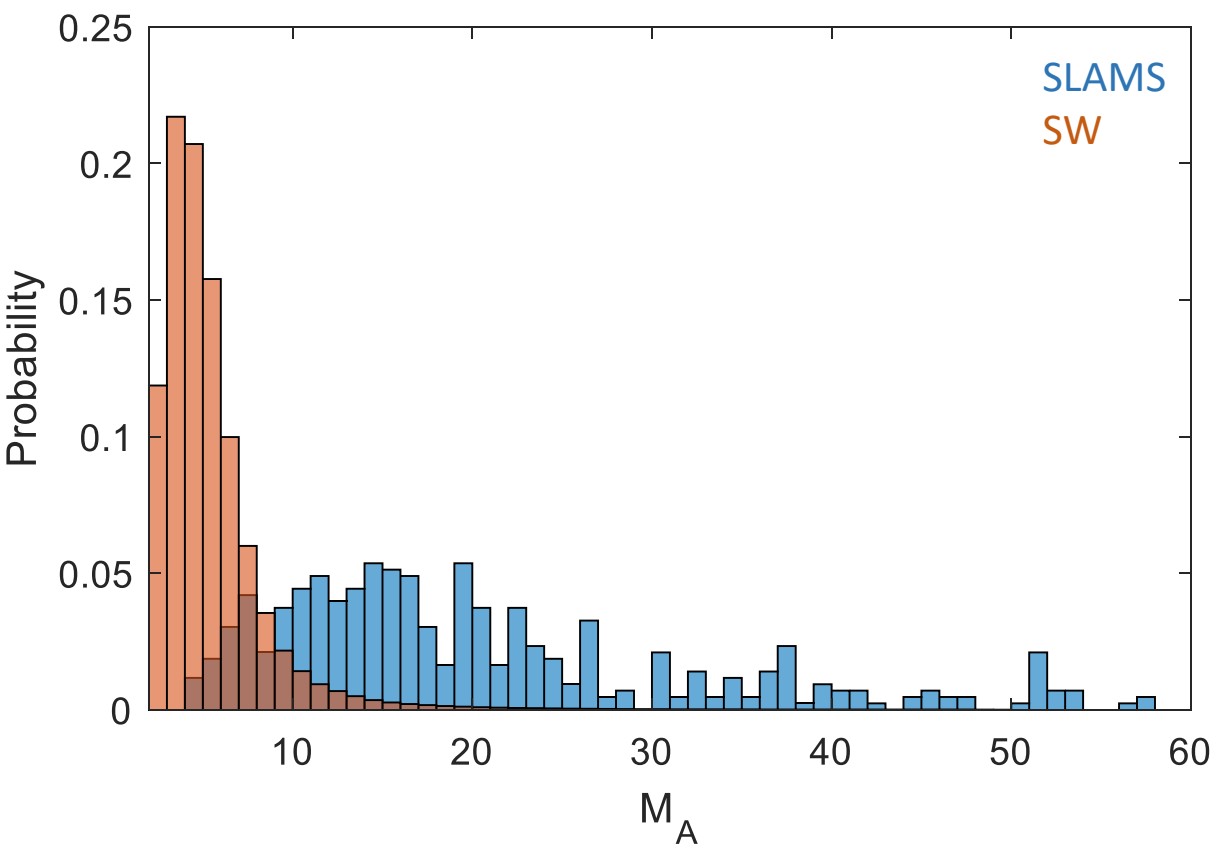

**Figure 7.** Distribution of Alfvén Mach number determined from the Tao solar wind model. Shown are Mach numbers for the whole MES-SENGER mission (red) and for the time intervals where SLAMS were observed (blue).

### 3.2.5 Relation to bow shock

Since SLAMS are expected to be strongly associated with the foreshock, we have investigated some aspects of positions of the SLAMS with respect to the bow shock. Figure 8 describes the geometry and defines the observational parameters. It is adapted from a similar description defined by Kajdič et al. (2017).

We first investigate if SLAMS are observed in the foreshock, which we here define in the broad sense that it is any point which is magnetically connected to the bow shock. The boundary of the foreshock, thus defined, is the last field line connected to the shock, which in Figure 8 is the line marked with $dB_t$ (which is the distance from the observational position and the connection point at the bow shock.) In order to determine if the observation point is connected to the shock, we follow the IMF field line starting from the SLAMS observation (determined by averaging over a 20 s window before and after the SLAMS),



**Table 2.** Number of SLAMS for each outcome of bow shock relation.

| Outcome | No. of SLAM |
| --- | --- |
| 0) SLAMS not connected to model bow shock | 53 |
| 1) SLAMS connected to model bow shock | 363 |
| 2) SLAMS inside model bow shock | 13 |

and determine if it connects to a model bow shock of the standard form (Winslow et al., 2013)

$$\sqrt{(x_{MSM} - x_0)^2 + y^2_{MSM} + z^2_{MSM}} = \frac{p\epsilon}{1 + \epsilon\cos\theta}. \tag{4}$$

For the focus point $x_0$, and the eccentricity $\epsilon$, we use the best-fit values of Winslow et al. (2013): $x_0 = 0.5$ ; $R_M, \epsilon = 1.04$, while we let the focal parameter $p$ vary. For each SLAMS event, we determine $p$ from the closest bow shock crossing, in the same way as was done by Glass et al. (2023). The values of $p$ so determined vary between 1.87 and 3.44 $R_M$, which can be

compared with the best fit value of Winslow et al. (2013) of $p = 2.75$ $R_M$. This process results in three possible outcomes: 0) the SLAMS position is not connected to the bow shock, 1) the SLAMS is connected to the bow shock, or 2) the SLAMS is located within the model bow shock. The latter case is a consequence of the uncertainties of the bow shock model. The number of each outcome is given in Table 2. We can see that a large majority of SLAMS are connected to the foreshock, or are located very close to the bow shock (as indicated by being inside the model shock).

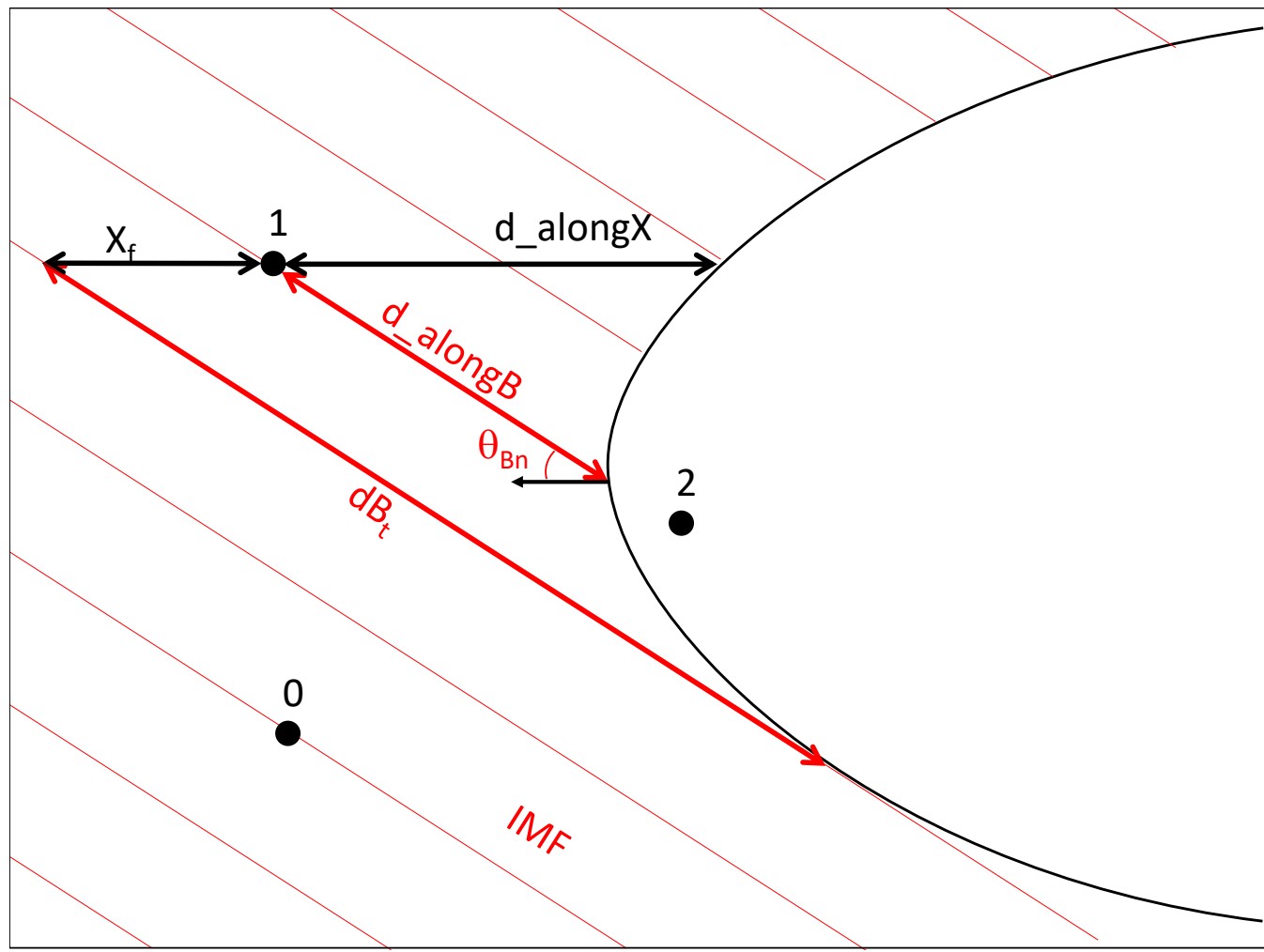

**Figure 8.** Definition of the foreshock coordinates, similar to those used by Kajdič et al. (2017)



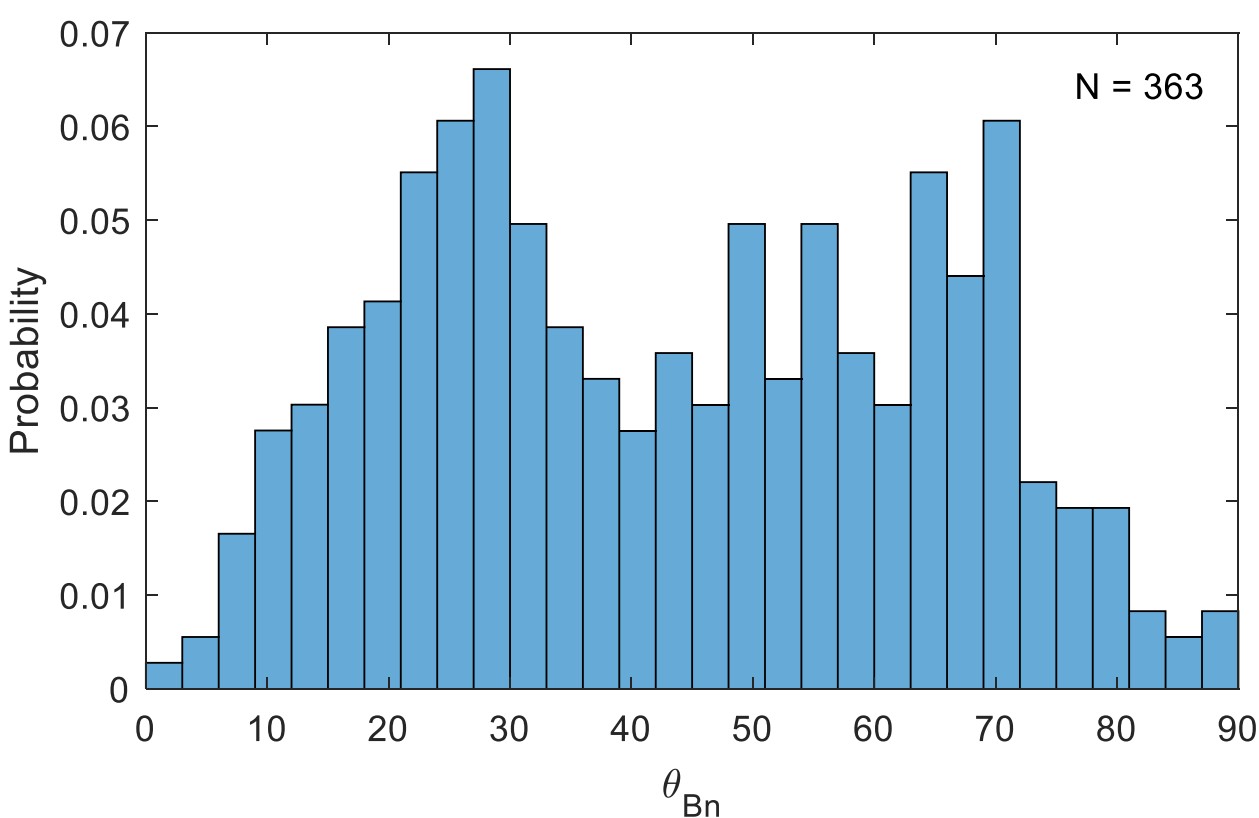

**Figure 9.** Distribution of $\theta_{Bn}$, the angle between the bow shock normal and the IMF.

For SLAMS connected to the bow shock, we evaluate the angle $\theta_{Bn}$ between the bow shock normal and the IMF at the point where the magnetic field line connects the SLAMS to the bow shock (see Figure 8). As can bee seen in Figure 9, there is no clear difference in SLAMS probability between the quasi-parallel and quasi-perpendicular configuration of the bow shock at the connecting point. This somewhat surprising result will be discussed later.

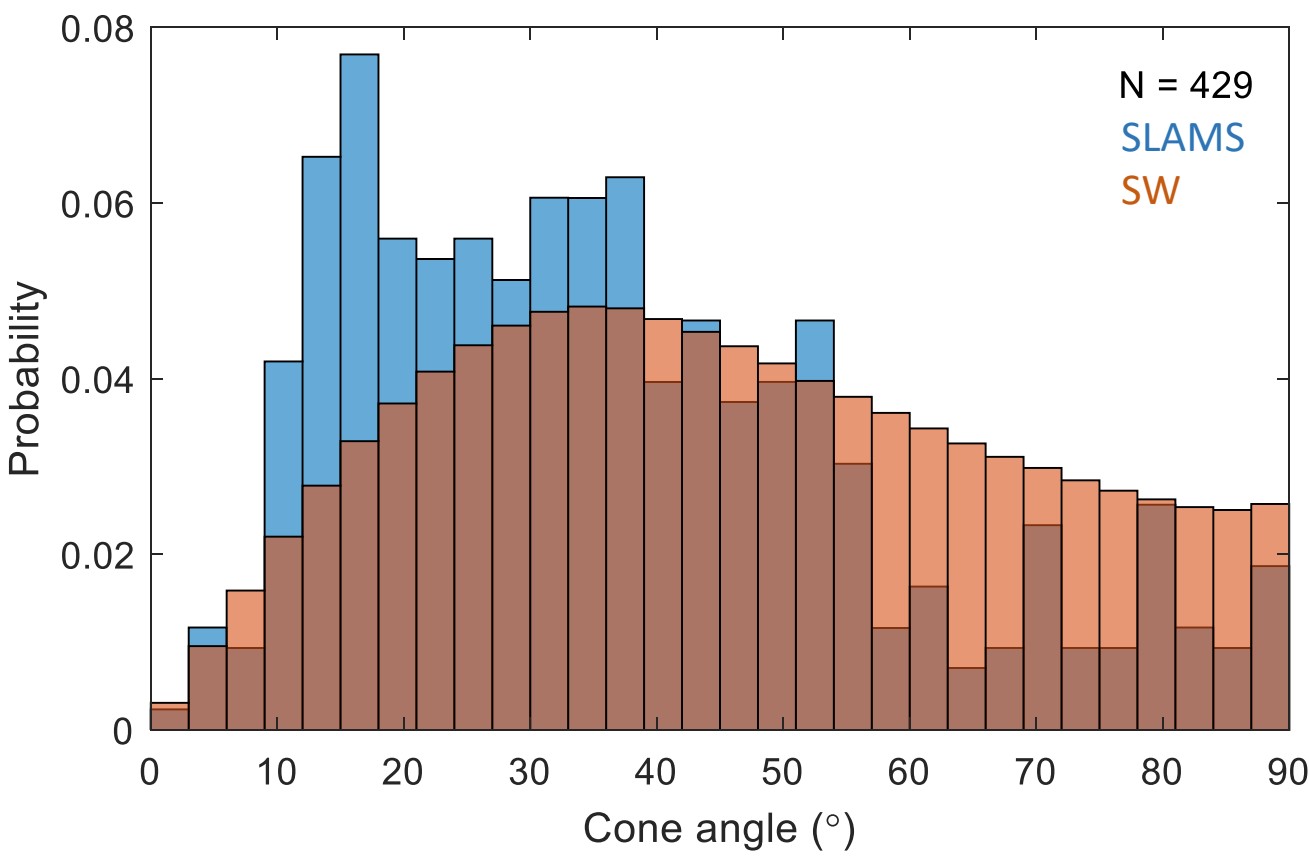

**Figure 10.** Distribution of solar wind magnetic field cone angle, both for SLAMS observation times (blus) and for the whole mission (red).

There is, however, a dependence on the IMF cone angle (defined as $\arctan(\frac{B_x}{\sqrt{B_y^2+B_z^2}})$), where the magnetic field corre-

sponding to each SLAMS again is determined by averaging the field before and after the SLAMS, as described above. The

blue distribution in Figure 10 is the cone angle distribution for SLAMS, while the red distribution is the cone angle distribution

for all solar wind measurements of the mission. As can be seen, SLAMS are generally found for smaller cone angles than the

general distribution.







**Figure 11.** Distributions of foreshock distance parameters. See text for definitions.



Finally, we show distributions of the distances defined in Figure 8. Panel (a) of Figure 11 shows the distribution of distance

to the bow shock along the $x_{MSM}$ axis ($d_{alongX}$). This distance is defined for all SLAMS that were not found inside the model bow shock, of which there are 416. It is clear that most SLAMS are found within $1\,\mathrm{R_M}$ (217 observations), and 310 events are found within $2\,\mathrm{R_M}$.

Similarly, the distance to the bow shock along the IMF, $d_{alongB}$, (defined for 363 events, and shown in panel (b)) is less than $1\,\mathrm{R_M}$ for a large majority of SLAMS (204), with 281 SLAMS within $2\,\mathrm{R_M}$.

Finally, the distance along the $x_{MSM}$ axis from the tangent field line, $X_f$ is shown in panel (c). The distribution is quite wide, reflecting that this distance depends on the cone angle and can become quite large for small cone angles. Positive values of $X_f$ correspond to positions downstream of the tangent field line, and it can be seen that the distribution extends to downstream distances of $13\,\mathrm{R_M}$. Negative values correspond to SLAMS upstream of the tangent field line. These are all found within $4\,\mathrm{R_M}$ of the tangent field line.

## 4  Discussion

The SLAMS as we have defined them in this study, have many properties that are similar to SLAMS at Earth; they are found on field lines connecting to the bow shock, they are short and rather isolated spikes in magnetic field strength, with a magnitude several times that of the background field strength. For a few cases we have also showed that they are elliptically, right-hand polarized, also consistent with terrestrial SLAMS (Schwartz, 1991).

Mercury SLAMS are most often found as individual large-amplitude signatures embedded in a general ULF wave field. This is consistent with the results of Schwartz et al. (1992), who state that SLAMS are often found within a 'surrounding well-developed ULF wave field'. Note that this type of SLAMS are called 'isolated' by Schwartz et al. (1992). Our category 'Wave package' type of SLAMS are likely to be a sub-category of this type, where the ULF foreshock boundaries (e.g. Le and Russell, 1992; Blanco-Cano et al., 1999, 2011) change on short time scales due to variations in the solar wind conditions,

and the wave package represents a brief excursion into the ULF foreshock region. With single-spacecraft measurements, it is difficult to verify this. Global kinetic simulations may help to address this question.

'Isolated' SLAMS in our nomenclature represent a phenomenon that has not been reported at the Earth foreshock. A possible explanation is that SLAMS propagate with a phase velocity which is different from both the ULF phase velocity (both in direction and magnitude), and the solar wind velocity. If they are created close to the ULF foreshock boundary, after a while they

may propagate out from the ULF foreshock region. A search for similar structures in Earth's foreshock, using Cluster multipoint measurements is on the future agenda. Regarding 'Boundary' SLAMS we can only offer some speculative explanations. One possibility is that they are not a really a separate class, but rather SLAMS that are found close to the ULF boundary, and the human propensity for classification is misguided in this case. Alternatively, they may be crossings of the so-called 'foreshock compressional boundary', suggested to sometimes coincide with the ULF wave boundary (Rojas-Castillo et al., 2013).

A further similarity to Earth SLAMS is that Mercury SLAMS sometimes exhibit whistler-like emissions at their upstream edge (Scholer et al., 2003; Wilson III et al., 2013). This type of whistler precursors are commonly observed at the quasi-





perpendicular bow shock (e.g. Walker et al., 1999), and is commonly taken to indicate that SLAMS can act as local quasi-perpendicular shocks (Tsubouchi and Lembège, 2004).

A majority of the SLAMS are found within 2 $R_M$ of the model bow shock, both as measure along the sun-planet line
$(d_{alongX})$, and along the IMF $(d_{alongB})$. This distance corresponds to the extent of what Schwartz and Burgess (1991) call the 'transition region'. No study of the size of the transition region at Earth exists, but Schwartz et al. (1992) speculate that it may be between 2 and 4 $R_E$, which means that the transition region at Mercury is considerably smaller in absolute terms. However, normalized to the size of the bow shock, which is around 20 times smaller than that of Earth (e.g. Russell, 1977), it is actually larger. An interesting comparison between the terrestrial and Hermean systems would be the ratio between $X_f$
and $d_{alongX}$, which says something about the growth rate of the non-linearity suggested to be responsible for the development of SLAMS from ULF waves. The relevant measure of growth rate is how many wave periods in the frame of the solar wind plasma are cycled through during the traversal of the distance $X_f$. Applying geometric similarity, and comparing the sizes of the transitions regions (using the mean value of $X_f$ of 4.8 $R_M$), we can estimate $X_f$ to be around 5 times greater than at Mercury (in absolute terms). Assuming the same solar wind velocity, and estimating the ULF wave frequency in the solar wind plasma to be proportional to the IMF (which is a very rough estimate), we see that the ratio of the plasma frame frequencies at Mercury and Earth is also around 5. Traversing the distance $X_f$ at Earth and Mercury, the plasma should therefore experience approximately the same number of wave periods, meaning that the growth rate associated with the non-linear development of SLAMS should be comparable between Earth and Mercury, once favourable conditions for SLAMS generation are present. This is a very rough estimate, obviously, and a closer comparison of growth rate estimates at different planets remains an important future subject to investigate.

The favourable conditions mentioned above should include the SLAMS field line to be connected to the quasi-parallel bow shock, and it is puzzling that no such trend can be see in Figure 10. However, it is known that properties characteristic of the quasi-parallel foreshock can extend into regions mapping to a $\theta_{Bn}$ of up to at least 60° (Bonifazi and Moreno, 1981; Karlsson et al., 2021b; Glass et al., 2023), especially during conditions of more radial IMF (Blanco-Cano et al., 2009). This is consistent with the drop-off of SLAMS observations for $\theta_{Bn} > 60°$, and the fact that SLAMS occur during times with a lower cone angle than the general solar wind distribution. An alternative explanation could be the small size of the Mercury bow shock. SLAMS at Mercury could have a spacial extent perpendicular to the IMF which may be a considerable fraction of an $R_M$. If SLAMS have a non-zero velocity in the solar wind plasma frame, their motion may result in parts of the SLAMS extending out into the more quasi-perpendicular foreshock.

Favourable conditions also seems to include a high Mach number. Scholer (1995) argues, based on simulations, that the density of diffuse ions (which are one type of the modified particle distributions mentioned in the Introduction) in the foreshock depends on the Alfvénic Mach number, and that this higher density gives rise to a higher probability of high-amplitude magnetic pulsations. It then seems likely that there is a minimum Mach number for SLAMS generation, all other things being equal. Since the Mach number is generally lower at Mercury than at Earth, this is consistent with SLAMS at Mercury being observed for higher Mach numbers than are typical at Mercury. Thus SLAMS should be more rare at Mercury than at Earth, and indeed they seem to be. MESSENGER spent around 15 000 h in the solar wind during the mission, and this corresponds to an observational



rate of around 0.03/hour, while Mandell (2020) reports on a rate of 1-5/hour at Earth. Thus the picture of the quasi-parallel bow shock being built up of SLAMS may be less relevant at Mercury than at Earth. For lower Mach numbers, instead the quasi-parallel bow shock may undergo a more global, cyclic reformation, as suggested by Sundberg et al. (2013). Further studies of

the dynamics of the quasi-parallel bow shock at Mercury will be an interesting subject for future BepiColombo studies.

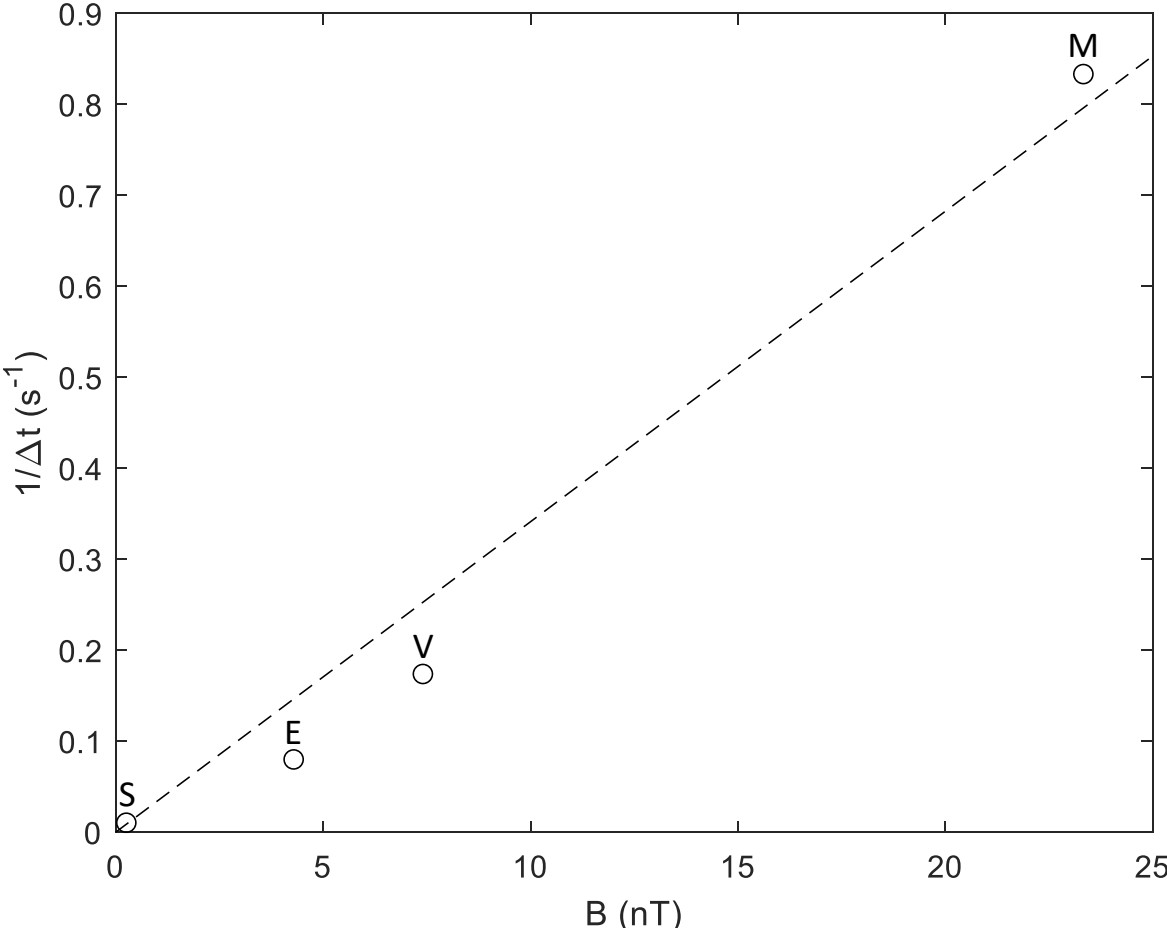

**Figure 12.** Typical inverse temporal scale sizes of SLAMS versus typical interplanetary magnetic field strenght at Saturn (S), Earth (E), Venus(V), and Mercury (M).

Finally, we can use the temporal scale sizes of Mercury SLAMS to investigate the connection to the terrestrial '30 s' ULF waves, and their analogues at other planets. Figure 12 shows the inverse temporal scales for SLAMS at Mercury, Venus, Earth, and Saturn, versus typical magnetic field strengths at these planets. For the temporal scale sizes, we have used the midpoint of the time scale intervals given for Venus, Earth, and Saturn, by Collinson et al. (2012), Schwartz et al. (1992), and Bebesi et al.

(2019), respectively. For Mercury we have used the mean value from the present investigation. For the magnetic field strength,





we have used the observations fits of the dependence of interplanetary magnetic field strength on heliocentric distance given by Behannon (1978).

It is known (Takahashi et al., 1984) that the frequency in the spacecraft frame of the '30 s'/ULF foreshock waves at Earth depends linearly on the interplanetary magnetic field strength ($B_{IMF}$). If this can be generalized to the other planets, and if

SLAMS grow by steepening of a half-period of ULF oscillations, SLAMS should inherit this dependence on $B_{IMF}$. Figure 12 confirms such a linear dependence. (Takahashi et al., 1984) give the observed ULF frequency as

$$f_{ULF} = 7.6 \cdot 10^6 B_{IMF}, \tag{5}$$

while a least squares fit to the observations in Figure 12 gives

$$\frac{1}{\Delta t} = 61 \cdot 10^6 B_{IMF}. \tag{6}$$

The factor eight difference between the inverse time scale and the ULF frequency is consistent with the steepening of ULF waves into more monolithic, isolated structures, as can be seen in Figure 1, and the 'Wave field' example of Figure 3. However, a more careful analysis with larger data sets from the other planets should be performed, since the observed frequencies depend not only on the strength of the IMF, but also its direction. Still, the comparison strongly supports that the SLAMS generation mechanism at different planets are similar, and are related to the properties of the original foreshock ULF wave field.

**5   Summary and conclusions**

We have performed a comprehensive search for SLAMS in the solar wind upstream of the Mercury bow shock, defined as increases of the magnetic magnitude of at least $\frac{\Delta B}{B_0} > 2$. 429 SLAMS were found for the whole mission.

The SLAMS found have some properties in common with those found at Earth; they are mostly found within a region of ULF waves, on field lines connecting to the bow shock, although not only for $\theta_{Bn} < 45°$. Other similarities to terrestrial SLAMS

are that, for a few examples, we have showed that SLAMS are right-handed, elliptically polarized (in the spacecraft frame), and that some SLAMS have a sharp upstream edge, and are sometimes associated with whistler-like emissions.

Other properties differ from SLAMS observed at Earth; SLAMS are likely to be much rarer at Mercury, which is consistent with an Alfvénic Mach number dependence. SLAMS are observed at Mercury for clearly higher Mach numbers than are typical at Mercury.We submit that this is important for the nature and dynamics of the quasi-parallel bow shock at Mercury.

The temporal scale sizes of SLAMS are smaller than those at Earth, which is consistent with the shorter periods of foreshock ULF waves at Mercury. This is support for the idea that SLAMS grow from the ULF waves in some type of non-linear interaction with the foreshock ion populations. A comparison with SLAMS found at Venus and Saturn lends further support to this scenario.

SLAMS at Mercury are found closer to the bow shock that at Earth, however, they may still be associated with similar

growth rates to terrestrial SLAMS.



While we have established the existence of SLAMS at Mercury, there are of course many unanswered questions. Many of these can be addressed by the upcoming BepiColombo mission, above all questions regarding the association to and interaction with different types of foreshock ion distributions.

*Author contributions.*

This study was conceived by TK and FP together. TK did the part of the data analysis that was connected to finding SLAMS and extracting their properties. FP was responsible for the methodology of the analysis of the connection to the bow shock, with contrubutions from TK and AG. JR and AG were responsible for providing the FIPS data. All authors contributed to formulating results and conclusions. The paper was mainly written by TK, with FP, AG, and JR helping with discussions and editing.

*Competing interests.*   No competing interests are present.

*Acknowledgements.*   TK and FP are grateful for the hospitality of the International Space Science Institute (ISSI), Bern. TK was supported by the Swedish National Space Agency grant no. 2020-00159.



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
