# Peer review of "Short Large-Amplitude Magnetic Structures (SLAMS) at Mercury observed by MESSENGER"

_Annales Geophysicae, 2023_

## Referee Comment (RC1)

Title: Short Large-Amplitude Magnetic Structures (SLAMS) at Mercury observed by MESSENGER

This paper deals with the topic of so-called SLAMS, which occur in the upstream region of the quasi-parallel bow shock, the foreshock, at Earth. The question if they also occur at Mercury is discussed, as an older study claimed a non-existence. Using a simple criterion to search for SLAMS in the solar wind upstream of Mercury in the magnetometer data from MESSENGER, the authors identify a total of 435 structures that can be labeled as SLAMS. A statistical study on the characteristics of the SLAMS is then performed and it is shown that there are, at least, 3 categories, maybe 6. It is shown that the structures are much shorter in time than those measured near Earth. They occur for lower-than-normal background magnetic field strength, and therefore, at higher-than-normal Alfvénic Mach number. There is a tentative result that the inverse duration of these SLAMS at different planets show a linear correlation between the average background magnetic field strength.

This paper is clearly written and gives an incentive for measurements with the upcoming orbital phase of the BepiColombo mission, with which the necessary plasma measurements will be made. There are only a few minor comments, which are listed below.

Comments:

- Line 80 and lines 105-108: "the Tao solar wind model"
  In the absence of good solar wind plasma data it is necessary to use some model in order to make calculations (e.g., for the Alfvén velocity). Many papers use the Tao model (I have done it myself). Most likely is for the back-tracing of the solar wind the error in the timing not too bad, but can the authors give an indication how well the model worked, for example by showing the magnetic field as measured and as propagated? This in order to check the timing. If I recall correctly, at comet 67P the error was several hours.
- Line 169: "The example shown in Figure 1 has a wave period of around 0.3 s, which can be compared to the approximate periods of 3-4 s, and 2 s for the 'Wave field', and 'Boundary' examples, respectively."
  I am not sure what the authors want to say here. Indeed, one can compare 0.3 s waves with the other values that are given. But what conclusion is one to get from this comparison? That the structures look alike? Maybe a few words more on what is meant here.
- Line 177: "SLAMS 170"
  This is probably a typo, or do the authors want to say something with the number 170?
- Lines 190-191: "it is anyway unlikely that the Mercury SLAMS would have the same dependence on the amplitude."
  Here I am confused, why would the SLAMS not have the same dependence? Later in the discussion section the authors are showing how the Hermean SLAMS and the Earth SLAMS are very much the same, albeit that the Hermean are much shorter.
- Figure 9: This almost looks like a double peaked distribution around ~30 and ~60 degrees. I am not sure if this could be anything significant, but it could have something to do with the nonlinear growth rate peaking around these angles. This might go too far to discuss in this paper, as it would need a full discussion of the dispersion and non-linear growth of waves.

Typos:

- Line 29: "that the are" → "that they are"
- Line 53: "large cross-section" → either "a large cross-section" or "large cross-sections"
- Line 60: "defines" → "define"
- Line 63: "uses" → "use"
- Line 133: "oscillations the middle" → "oscillations in the middle"
- Line 142: 'has has"
- Line 162: "which do not DeltaB/B > 2", there is a verb missing here
- Lines 174-175: this sentence has twice "may", delete the one in line 175.
- Line 227: "x0 = 0.5 ; Rm" probably take out ";"

---

## Referee Comment (RC2)

**Review report on "Short Large-Amplitude Magnetic Structures (SLAMS) at Mercury observed by MESSENGER" by T. Karlsson, F. Plaschke, A.N. Glass, and J.M. Raines.**

This work is a valuable investigation of SLAMS at Mercury. The authors surveyed 4 years of MESSENGER data finding 429 events identified as SLAMS in the near-Mercury environment for later categorization into 4 types according to specific characteristics of the events. This is already an important achievement as previous studies of SLAMS in Mercury have a case-by-case approach. Statistics on duration, location, and magnetic field are presented in this work, then authors move to explore possible dependencies of SLAMS, particularly in terms of the geometry of the shock and their relation to this structure. Though some questions arise in the first part of the manuscript which would benefit from some clarifications, the second part is in need of more discussion.

The authors explore the full near-Mercury environment and found SLAMS-like signatures, including both quasi-parallel and perpendicular upstream regions. There is a problem from the beginning, as the authors do not confine the search to the foreshock region where ULF waves exist and can evolve into SLAMS. When considering the quasi-perpendicular region the authors found events that resemble SLAMS' features but is hard to think they truly are such structures as there is no source for them there. The are two main concerns about this that the authors do not succeed to explain/sustain and even contradict themselves (see my comments on Sec. 3.2.5 and on lines 305-313 of the Discussion):
- SLAMS-like events are found in regions with large $\theta_{Bn}$, which is unexpected as such structures are native to the quasi-parallel region upstream of the shock.
- SLAMS-like events are found in regions magnetically unconnected to the bow shock, despite such structures are related to the foreshock.

It is well written with a logical order in the presentation of results and discussion. Being a statistical analysis it gives good information on these structures and potentially will allow to establish comparison of such structures at Mercury and other planetary environment. However, as said before some of the physical interpretation needs stronger arguments/reasons, and so the reviewer asks for major revisions before the manuscript could be in shape for publication.

**MAJOR COMMENTS**

Lines 5-6: The author mentioned studying cases of SLAMS isolated from ULF waves which differ substantially from the definition of SLAMS as steepened waves. ¿What is the SLAMS formation mechanism in that case?

Lines 15, 20: The foreshock is, by definition, the region magnetically connected to the shock as it is the region where SW incident particles are reflected by the shock meaning that is the quasi-parallel region upstream of the shock. But here it says it can be connected to the quasi-perpendicular portion which is not true.

Line 50: What does it mean cyclic behavior of SLAMS?

Sec. 2.3 Does the Tao model work only for quiet solar wind conditions? Is this the best model to use when lacking plasma measurements? Could ENLIL work best for this purpose?

Line 120: Though the enhanced particle flux and waves are a good sign of being in the foreshock region, the magnetic field depression around 11:10 UT is not consistent. We would expect strong perturbations of the field but not such a decrease. Even more, the field magnitude goes well below the ambient solar wind or even the values for the foreshock at around 10:45 UT. One even could argue that this structure resembles a Hot Flow Anomaly.
i) What is behaving the ion velocity?
ii) What is the authors' physical interpretation of such low B-field?
iii) How far from the bow shock is the SLAMS located?
iv) Is the event here presented the most typical one in the catalog? Is it representative of the other 428 cases?

Sec. 3.1 The zoom-in of the event (Figure 1h-j) shows a nice analogy with SLAMS observed at Earth as the authors pointed out (shape, compression, polarization, and frequency). But the region where these structures are seems to be more complex than just a foreshock region (Figure 1 a-d). I would suggest including a more "typical" (Earth-like) case and keeping this complex event with more discussion on the plasma surrounding the SLAMS.

Line 147-149: Could the authors check for the solar wind velocity or dynamic pressure of all the cases, particularly those within the sheath? To have an idea whether they were observed during periods of fast/dense solar wind or even during the passage of some solar transient so that the bow shock location could be moved closer to the planet. Perhaps this would also give information on SLAMS and solar wind dependency.
i) How many SLAMS are downstream of the modeled bow shock?
ii) There are two clear outliers in the sample, around $x \sim 2.2 R_M$ and $rho \sim 4.3 R_M$. Do they have any particular signature or difference from the rest?

Sec. 3.2.1 a) Are the types of SLAMS somewhat related to the distance they were observed? Can they be identified as clusters in Figure 2.
b) The "higher frequency" category seems to be just very compressive waves that satisfy the only B-field criteria. In that sense, there is a reasonable doubt that the 40 structures in this category are SLAMS. Do the authors check if they are embedded in the foreshock region (ion spectra) in the visual inspection? What is the angle $\theta_{Bn}$ for those events?

Sec. 3.2.2 a) Are the 11 whistler precursors observations the only ones in the whole dataset? Or just 11 cases were labeled as such? b) Are the 11 cases within the "sharp" SLAMS category? c) I suggest including an event positively identified as a SLAMS instead of the shocklet presented in this section for continuity of the main topic which is SLAMS.

Sec. 3.2.3 From Figure 3 it is clear that isolated/sharp structures will have shorter/larger duration than any other category, possibly infringing some biased in the statistics. On the other hand, the duration reported by Schwartz et al. (1992) corresponds to SLAMS of the wave field or sharp category. In order to have a better comparison with their terrestrial counterparts, it would be good to include the statistics for each category.

Sec. 3.2.4 The solar wind around Mercury is characterized by a low Alfvénic mach number, typically 4-6 but it can be much lower.
a) In Figure 6 the $B_o$ for SW periods peaks at 20 nT, which is far from expected. Is this discrepancy only due to the errors in Tao's model? What else could be responsible for it?
b) From Figure 6 it is true that as a global trend, the SLAMS will still be more prone to appear during very low $M_A$, which can be satisfied for a faster/denser solar wind or a low B-field. The authors focus on the second case, but no discussion on the first case is mentioned possibly because V and N are model derived. The manuscript will benefit from more discussion on how to overcome the FPI limitations and also the pros and cons of using Tao's model.
c) Romanelli et al., 2021 showed that the occurrence of ULF waves increases for low B-field for heliocentric distances between 0.31 and 0.47 AU, which is associated with a larger reflection of SW protons as the heliocentric distance increases. The reflected particles work as the source of such waves. This will actually also explain why in the present study SLAMS defined as the result of the non-linear evolution of ULF waves, are observed for low $B_o$ values.
d) A reference to the SW $M_A$ should be included. See e.g. Romanelli et al. (2021) and references therein.

Sec . 3.2.5
a) How many events are downstream of the nominal bow shock in Figure 1? Is it a similar number to the groups 0 and 2 in Table 2?
b) Line 235: How was calculated the angle? Indeed this is a surprising result as SLAMS would be related to the foreshock (quasi-parallel) region where the ULF waves exist.
c) If a SLAMS is connected to the shock, that means the structure is upstream of the quasi-parallel shock. Then, according to Table 2, those are 363 events and so 66 events are in the quasi-perpendicular portion of the shock. But when calculating $\theta_{Bn}$, Figure 9, the probability of SLAMS in the q-perpendicular region is not much lower than for the q-parallel region. These two results contradict each other.
d) One could argue that those found in the quasi-perpendicular region where not originated there but rather in the quasi-parallel region and later traveled to the other region. Actually, the authors make this point for the "isolated" SLAMS category, which sounds reasonable and could also explain -at least in part- why in Figure 9 the $\theta_{Bn}$ is so broad.
e) Line 244-249: the middle panel of Figure 11 includes only the connected events, while the other panels include connected and not connected which prevents the reader from a direct comparison. The authors should show all panels for only the connected events. See my comment on lines 256-257.
f) Line 250: This result again sustains the fact that the probability of SLAMS in the foreshock region (connected to the shock, Table 2) is higher, but opposite to what is shown in Figure 7. This review has no other but to ask again for a response to such a dichotomy.

Line 256-257: The authors note the SLAMS they studied -similar to Earth- "are found on field lines connecting to the bow shock". Therefore SLAMS are foreshock structures and so they are 363 of such SLAMS which should have $\theta_{Bn}$ angles typical for the quasi-parallel region upstream of the shock. The problem explained in my comments on Sec. 3.2.5 prevails here.

Lines 267-274: What are the $\theta_{Bn}$ angles of the 6 isolated events? Are they in the "connected to the shock" group? This reviewer thinks the connectivity to the shock would be more important to check before the type of structures, as it will be warranted that the structures are in the foreshock. Could the authors check for the 363 events connected to the bow shock how many structures are for each category? Is the isolated category still there?

Lines 284-286: How this ratio will give information on the wave growth rate? Please explain.

Lines 287-295: One can argue that because of the weak bow shock at Mercury, in terms of its $M_A$, the reflection of particles is less effective than at Earth. This would mean that waves possibly need more time to develop from the resonance of incident and backstreaming particles. The number of wave periods for a "geometrically equivalent" distance, and hence growth rate, would be lower at Mercury than at Earth.
On the other hand, authors could find some references to the typical ULF waves frequency at Earth and Mercury, calculate the ratio and on the other hand calculate the number of wave periods separately for each planet. This way they could avoid "rough estimates" and have more realistic numbers.

Lines 297-301: In the cited works, indeed it is shown that some characteristics typical of a foreshock region can be found for angles $\theta_{Bn}$ larger than 45°; however it is very clear that the distribution of parameters peaks for 15-40° (see Glass et al., 2023). Events beyond $\theta_{Bn}$=50° are scarce and do not representative of the whole sample. This scenario differs widely from what is presented here in Figure 9 and cannot be compared. What is more, Blanco-Cano et al., 2009 reported ULF waves far from the nose of the shock for a radial IMF configuration and for low cone angle, $\theta_{BV}$, that is still well within the foreshock; but do not report foreshock-like parameters/particles/waves for large $\theta_{Bn}$. In brief, the explanation for the unexpected behavior of $\theta_{Bn}$ in Figure 9 is reasonably arguable.

Lines 301-304: The authors could estimate the size of the SLAMS along the spacecraft trajectory, either using a proxy (case by case) or an average of the $V_{sw}$. Normalized to $R_m$ would give a better idea of how extended are the structures in order to help sustain this alternative explanation.
Actually, one can take the 1.2s with an average $V_{SW}$~460 km/s (Diego et al. 2020 and references therein), we found the extension to be $0.22R_M$. This reviewer finds it hard to think that a SLAMS with a cross-section of (at least) 0.02 $R_M$ located in the quasi-parallel region could be extended into the quasi-perpendicular shock. At most, one could think that if this SLAM is located at the edge of the foreshock then the statement is true; but according to the statistical results in Figure 9, there would be a very large number of SLAMS with this configuration.

Lines 305-313: In lines 286-293 and after some rough estimations on the wave growth rate in Mercury and Earth foreshocks, the authors conclude that "the non-linear development of SLAMS should be comparable between Earth and Mercury". Such result go against the new interpretation presented now in the discussion section.

Line 338-339: As mentioned before this conclusion is reasonably arguable and should be adapted after a detailed revision of all the reviewer comments.

MINOR COMMENTS
Lines 25: the → then
Line 111: Should said "Example" as only one case is presented.
Line 133: "higher frequency oscillations **IN** the middle"
Line 142: "… has "
Line 177: "SLAMS 170" possibly a typo?
Line 180: "gyro radius" → gyrofrequency
Line 200: On → One
Line 227: $0.5;R_M$ → $0.5R_M$
Figure 6: The unit for $B_o$ is missing in the label for the horizontal axis.

References:

1. Romanelli, N., DiBraccio, G.A. Occurrence rate of ultra-low frequency waves in the foreshock of Mercury increases with heliocentric distance. (2021) Nat. Commun. 12, 6748. https://doi.org/10.1038/s41467-021-26344-2
2. Zhang, H., Zong, Q., Connor, H. et al. (2022) Dayside Transient Phenomena and Their Impact on the Magnetosphere and Ionosphere. Space Sci Rev 218, 40. https://doi.org/10.1007/s11214-021-00865-0
3. Diego, P., Piersanti, M., Laurenza, M., & Villante, U. (2020). Properties of solar wind structures at Mercury's orbit. Journal of Geophysical Research: Space Physics, 125, e2020JA028281. https://doi.org/10.1029/2020JA028281

---

## Author Comment (AC1)

**Reply, Reviewer 1**

We are grateful to the reviewer for a careful reading of the manuscript. We give point-to-point replies below. Our answers to the original Reviewer comments are given in italic.

Line 80 and lines 105-108: "the Tao solar wind model" In the absence of good solar wind plasma data it is necessary to use some model in order to make calculations (e.g., for the Alfvén velocity). Many papers use the Tao model (I have done it myself). Most likely is for the back-tracing of the solar wind the error in the timing not too bad, but can the authors give an indication how well the model worked, for example by showing the magnetic field as measured and as propagated? This in order to check the timing. If I recall correctly, at comet 67P the error was several hours.

*The Tao model data available to us only have a time resolution of 10 min, so already there is a large uncertainty in the timing, likely comparable to the timing error. Any statistical results from the Tao model should therefore be taken with considerable caution anyway, and any correlations are probably only valid on long time scales. The stronger evidence for a connection between SLAMS and a high Mach number is the measured low background magnetic field strength. The model calculations of the Mach number should only be seen as additional circumstantial evidence, and a proper measurements of the Mach number will have to wait for the BepiColombo mission. We will add a discussion to the paper along these lines.*

Line 169: "The example shown in Figure 1 has a wave period of around 0.3 s, which can be compared to the approximate periods of 3-4 s, and 2 s for the 'Wave field', and 'Boundary' examples, respectively." I am not sure what the authors want to say here. Indeed, one can compare 0.3 s waves with the other values that are given. But what conclusion is one to get from this comparison? That the structures look alike? Maybe a few words more on what is meant here.

*For the 'Higher frequency' type, we have made note of several events that fulfilled the SLAMS search criterion, but seem to be steepened waves with frequencies not ordinarily associated with the Mercury analogs to terrestrial 30-s waves. As we pointed out in line 93, these analogs typically have periods of 1-20 s. We here point out that the frequencies of 'Wave field', and 'Boundary' types are consistent with this, and that higher-frequency events may warrant their own sub-group. It is possible that they are associated with Mercury analogs to terrestrial 3 s waves [e.g. Blanco-Cano, et al., 1999], rather than the 30 s waves. We will add a short discussion on this.*

Line 177: "SLAMS 170" This is probably a typo, or do the authors want to say something with the number 170?

*Yes, this is a typo.*

Lines 190-191: "it is anyway unlikely that the Mercury SLAMS would have the same dependence on the amplitude." Here I am confused, why would the SLAMS not have the same dependence? Later in the discussion section the authors are showing how the Hermean SLAMS and the Earth SLAMS are very much the same, albeit that the Hermean are much shorter.

*We meant that the detailed dependence of velocity in the solar wind frame on amplitude is unlikely to be the same as at Earth, since the plasma parameters at the two planets are rather different. It is therefore difficult to know how the results of Mann et al. [1994] can be used to convert temporal scale sizes at Mercury to spatial ones. We will clarify this.*

Figure 9: This almost looks like a double peaked distribution around ~30 and ~60 degrees. I am not sure if this could be anything significant, but it could have something to do with the non-linear growth rate peaking around these angles. This might go too far to discuss in this paper, as it would need a full discussion of the dispersion and non-linear growth of waves.

*It is difficult to know if the tendency for a double peak is significant. We will mention the possibility in the discussion, but will point out that this is speculation, and needs verification with further statistics, hopefully available with BepiColombo.*

**Typos**

*We will take note of these in the revision.*

*Note that in line 296, 'Figure 10' should be changed to 'Figure 9'.*

**References**

Blanco-Cano, X., Le, G., & Russell, C. T. (1999). Identification of foreshock waves with 3-s periods. Journal of Geophysical Research: Space Physics, 104(A3), 4643-4656.

Mann, G., Lühr, H., and Baumjohann, W.: Statistical analysis of short large-amplitude magnetic field structures in the vicinity of the quasiparallel bow shock, Journal of Geophysical Research: Space Physics, 99, 13 315–13 323, 1994.

---

## Author Comment (AC2)

**Reply, Reviewer 2**

We are grateful to the reviewer for a careful reading of the manuscript. We give point-to-point replies below, in italic.

**General remarks**

The authors explore the full near-Mercury environment and found SLAMS-like signatures, including both quasi-parallel and perpendicular upstream regions. There is a problem from the beginning, as the authors do not confine the search to the foreshock region where ULF waves exist and can evolve into SLAMS.

*We believe that it would be poor scientific methodology to only look for SLAMS where our preconceived notions tells us there should be. It is better methodology to first look for SLAMS (according to some definition), and the try to understand why they are observed where they are.*

When considering the quasi-perpendicular region the authors found events that resemble SLAMS' features but is hard to think they truly are such structures as there
is no source for them there. The are two main concerns about this that the authors do not succeed to explain/sustain and even contradict themselves (see my comments on Sec. 3.2.5 and on lines 305-313 of the Discussion):

SLAMS-like events are found in regions with large θBn, which is unexpected as such structures are native to the quasi-parallel region upstream of the shock.

*Again, we agree that it is unexpected, but on the other hand there may be differences between the situation at Mercury as compared to what we expect. We are indeed not aware of any study of the ULF foreshock boundary at Mercury (see below). We plan to do such a study as a floowo-up. We have also discussed possible reasons for these observations below.*

SLAMS-like events are found in regions magnetically unconnected to the bow shock, despite such structures are related to the foreshock.

*As discussed below, such cases are rare and can have several reasons, the main probalbly being uncertainties in the bow shock model.*

**Major comments**

*Lines 5-6: The author mentioned studying cases of SLAMS isolated from ULF waves which differ substantially from the definition of SLAMS as steepened waves. What is the SLAMS formation mechanism in that case?*

We have suggested a possible formation mechanism in the Discussion, lines 267-270.

*Lines 15, 20: The foreshock is, by definition, the region magnetically connected to the shock as it is the region where SW incident particles are reflected by the shock meaning that is the quasi-parallel region upstream of the shock. But here it says it can be connected to the quasi-perpendicular portion which is not true.*

Different authors use different definitions of the foreshock. The definition used by us in this study is consistent with the following:

Le et al., [2013]: 'The foreshock is the spatially asymmetric region magnetically connected to the planetary bow shock.'

Romanelli et al. [2020]: 'The foreshock is the spatial region upstream of, but magnetically connected to the bow shock.'

Romanelli et al., [2021]: 'A planetary foreshock is the spatial region upstream of, but magnetically connected to, a planet's bow shock.'

Kis et al. [2007]: 'The foreshock is the upstream region which is magnetically connected to the bow shock and is dominated by waves and energized particles. … The field-aligned beam (FAB), composed of collimated ion beams with an energy of a few keV, propagates along the interplanetary magnetic field away from the bow shock … and originates at regions of the bow shock where $\theta_{Bn}$ is between 45° and 70° i.e. at the quasi-perpendicular side.'

Kis et al. define the 'deep foreshock region as the quasi-parallel part of the foreshock: 'focusing on the processes in the deep foreshock region, i.e. on the quasi-parallel side.'

Eastwood et al. [2005]: 'The region of space upstream of the bow shock, magnetically connected to the shock and filled with particles backstreaming from the shock is known as the foreshock. … For the quasi-perpendicular bow shock ($\theta_{Bn} > 45$), the foreshock is restricted to the shock foot, while in the quasi-parallel part of the bow shock (($\theta_{Bn} < 45$) covers a much larger upstream domain'

[Figure]

*[Figure from Eastwood et al., 2005]*

The definition hinted at by the reviewer is more consistent with what Eastwood et al. call the ULF foreshock (see figure), which by the Eastwood definition is an admittedly important part of the foreshock, but not the whole foreshock. We will comment on our definition of the foreshock in the revised manuscript.

*Line 50: What does it mean cyclic behavior of SLAMS?*

Sundberg et al. [2013] write 'The stable frequency observed in the magnetosphere indicates that the subsolar magnetosphere was primarily influenced by a cyclic shock reconfiguration,

rather than a patchwork of quasi-simultaneous SLAMS, as is the general case at the terrestrial bow shock.' We will reformulate explaining that we mean a periodic behavior due to cyclic reformation.

Sec. 2.3 Does the Tao model work only for quiet solar wind conditions? Is this the best model to use when lacking plasma measurements? Could ENLIL work best for this purpose?

*We did also try to use the ENLIL model with very similar results, but decided to use the Tao model due to the higher time resolution in the model data available to us. Note that we only use the solar wind model data to calculate the Mach number, which we use to validate the purely data-based result of Figure 6 (typically lower magnetic field strength for SLAMS). We do point out that solar wind modelling of course can have large uncertainties, and will point out that this only serves as a preliminary validation to be tested with particle data from the upcoming BepiColombo mission.*

Line 120: Though the enhanced particle flux and waves are a good sign of being in the foreshock region, the magnetic field depression around 11:10 UT is not consistent.

*It is not clear why a magnetic field depression is inconsistent with a foreshock region. It is clear that it is mainly the x component of the magnetic field that changes, and therefore the direction of the IMF changes, affecting the relation to the bow shock normal in a temporal sense.*

We would expect strong perturbations of the field but not such a decrease. Even more, the field magnitude goes well below the ambient solar wind or even the values for the foreshock at around 10:45 UT. One even could argue that this structure resembles a Hot Flow Anomaly.

*This structure is not likely to be an HFA, the time scales of which have been reported to be of the order of tens of seconds [Uritsky et al., 2014], while this structure is of order of ten minutes.*

i) What is behaving the ion velocity?

*Due to the limited view of FIPS, plasma moment calculation is not possible.*

ii) What is the authors' physical interpretation of such low B-field?

*This may be a short excursion into the heliospheric current sheet. This is consistent with the variation being mainly in Bx, and such excursions may have time scales from minutes to hours [Szabo et al., 2020]. The structure may also be relate4 to the solar wind 'magnetic decreases' reported by Tsurutani et al. [2005],*

iii) How far from the bow shock is the SLAMS located?

*This event is located at $r_{MSO}$ = (-1.4,-0.1,-4.7) $R_M$, which is close to the bow shock (compare Figure 2). Referring to Figure 8, d_alongX for this event is 1.15 $R_M$.*

iv) Is the event here presented the most typical one in the catalog? Is it representative of the other 428 cases?

*It is quite typical, in that it is seen for a low value of B, which this event nicely exemplifies, and it is also typical in its temporal scale size and polarization properties, and being observed within a general ULF wave field.*

Sec. 3.1 The zoom-in of the event (Figure 1h-j) shows a nice analogy with SLAMS observed at Earth as the authors pointed out (shape, compression, polarization, and frequency). But the region where these structures are seems to be more complex than just a foreshock region (Figure 1 a-d). I would suggest including a more "typical" (Earth-like) case and keeping this complex event with more discussion on the plasma surrounding the SLAMS.

*As discussed above, this event nicely illustrates SLAMS observed in a clearly delineated region of foreshock-like ions, weak background magnetic field, and a background of compressional wave activity. We therefore think it is a good illustration of when SLAMS are observed at Mercury.*

Line 147-149: Could the authors check for the solar wind velocity or dynamic pressure of all the cases, particularly those within the sheath? To have an idea whether they were observed during periods of fast/dense solar wind or even during the passage of some solar transient so that the bow shock location could be moved closer to the planet. Perhaps this would also give information on SLAMS and solar wind dependency.

*Note that we do not believe that any SLAMS are found in the magnetosheath, the SLAMS behind the model bow shock we believe are still in the solar wind, it is ratherdue to uncertainties in the bow shock mode, which we point out in line 232. Due to the uncertainties of the solar wind models, we believe that a closer investigation with dependency on solar wind plasma parameters will have to wait for the BepiColombo mission.*

i) How many SLAMS are downstream of the modeled bow shock?

*13, as stated in Table 2,*

ii) There are two clear outliers in the sample, around x~2.2RM and rho~4.3RM. Do they have any particular signature or difference from the rest?

*No, these are typical events, classified as the 'Wavefield' type.*

Sec. 3.2.1 a) Are the types of SLAMS somewhat related to the distance they were observed? Can they be identified as clusters in Figure 2.

*No, we can see no clear trend in this respect. But the sample size of the types which are not 'W' or 'H' is rather small.*

b) The "higher frequency" category seems to be just very compressive waves that satisfy the only B-field criteria. In that sense, there is a reasonable doubt that the 40 structures in this category are SLAMS. Do the authors check if they are embedded in the foreshock region (ion spectra) in the visual inspection? What is the angle θBn for those events?

*We have not made a visual inspection of the particle data. The events have a rather flat distribution in qBn between 0 and 70°, with an average of 48°. We agree that it is a matter of definition if we should call them SLAMS, but since they fulfill the simple definition of SLAMS*

*that we have used, we feel that that we should not exclude them, rather mention that they are a subcategory which may or may not be closely related to the other types. This is a matter for further investigation in the future.*

Sec. 3.2.2 a) Are the 11 whistler precursors observations the only ones in the whole dataset? Or just 11 cases were labeled as such?

*These are the only ones where we found a clear whistler signature. The temporal resolution is, however, probably affecting the number of such observations.*

b) Are the 11 cases within the "sharp" SLAMS category?

*No, only one of these were classified as 'sharp'. (This can easily be seen in the supporting dataset.)*

c) I suggest including an event positively identified as a SLAMS instead of the shocklet presented in this section for continuity of the main topic which is SLAMS.

*The second example in Figure 4 was classified as a SLAMS. We find it interesting and worth reporting that also at Mercury shocklets with whistler signatures can be observed.*

Sec. 3.2.3 From Figure 3 it is clear that isolated/sharp structures will have shorter/larger duration than any other category, possibly infringing some biased in the statistics. On the other hand, the duration reported by Schwartz et al. (1992) corresponds to SLAMS of the wave field or sharp category. In order to have a better comparison with their terrestrial counterparts, it would be good to include the statistics for each category.

*We cannot agree that this is clear from Figure 3. From Figure 5, it is however clear that all subcategories have temporal scale sizes of less than 5 s, clearly shorter than those of Schwartz et al (1992). However, we have calculated the average scale sizes for the different categories, summarized below. We will add these to the revised manuscript. It is clear that the H category has a shorter scale size, but does not strongly affect the overall statistics.*

| Type | Avg. scale size (s) |
|---|---|
| W | 1.29 |
| B | 1.09 |
| P | 0.75 |
| I | 1.51 |
| S | 1.17 |
| H | 0.40 |
| All | 1.18 |
| All, except H | 1.26 |

Sec. 3.2.4 The solar wind around Mercury is characterized by a low Alfvénic mach number, typically 4-6 but it can be much lower.

a) In Figure 6 the $B_o$ for SW periods peaks at 20 nT, which is far from expected. Is this discrepancy only due to the errors in Tao's model? What else could be responsible for it?

*The results in Figure 6 are solely based on MESSENGER data and are nicely consistent with the results in [Hanneson et al., 2020], their Figure 2.*

b) From Figure 6 it is true that as a global trend, the SLAMS will still be more prone to appear during very low MA, which can be satisfied for a faster/denser solar wind or a low B-field. The authors focus on the second case, but no discussion on the first case is mentioned possibly because V and N are model derived. The manuscript will benefit from more discussion on how to overcome the FPI limitations and also the pros and cons of using Tao's model.

*We focus on the first case, since the magnetic field measurements are the only reliable indications of MA using only MESSENGER data. Due to the limited viewing angle of the FIPS instrument, routine density and velocity measurements are unavailable in the solar wind. The general case is of including density and velocity we try to verify by using Tao mdel data, as discussed above (see Figure 7).*

c) Romanelli et al., 2021 showed that the occurrence of ULF waves increases for low B-field for heliocentric distances between 0.31 and 0.47 AU, which is associated with a larger reflection of SW protons as the heliocentric distance increases. The reflected particles work as the source of such waves. This will actually also explain why in the present study SLAMS defined as the result of the non-linear evolution of ULF waves, are observed for low Bo values.

*We agree, and this is in general consistent with in increased amount of reflected particles for higher Mach numbers [e.g. Romanelli and DiBraccio, 2021], which are again consistent with lower magnetic fields. We will add a comment on how the Mach number and amount of reflected particles are related.*

d) A reference to the SW MA should be included. See e.g. Romanelli et al. (2021) and references therein.

*We will add a reference for earlier results on typical Mach numbers at 0.3 AU, in connection with Figure 7. Diego et al. [2020], for example, give typical values between about 2 and 10, consistent with our Figure 7.*

Sec . 3.2.5
   a) How many events are downstream of the nominal bow shock in Figure 1? Is it a similar number to the groups 0 and 2 in Table 2?

*We assume the reviewer means Figure 1. Here is only shown the bow shock for nominal solar wind conditions (only meant for visual help), while Table 2 is based on a bow shock model fitted to the last bow shock crossing. Figure 2 is therefore not directly comparable with Table 2.*

b) Line 235: How was calculated the angle? Indeed this is a surprising result as SLAMS would be related to the foreshock (quasi-parallel) region where the ULF waves exist.

*As described in lines 235-236: 'For SLAMS connected to the bow shock, we evaluate the angle $\theta_{Bn}$ between the bow shock normal and the IMF at the point where the magnetic field line connects the SLAMS to the bow shock'. The normal was calculated analytically from the bow shock model.*

c)  If a SLAMS is connected to the shock, that means the structure is upstream of the quasi-parallel shock.

*No, this is incorrect. The last field line connecting to the bow shock, the tangential field line, has a θ$_{Bn}$ of 90°, as can easily be seen in the Figure from [Eastman et al., 2005] above. As you go deeper into the foreshock, as we have defined it, the angle decreases until you reach the quasi-parallel region, which approximately coincides with the ULF foreshock region.*

Then, according to Table 2, those are 363 events and so 66 events are in the quasi-perpendicular portion of the shock. But when calculating θBn, Figure 9, the probability of SLAMS in the q-perpendicular region is not much lower than for the q-parallel region.
These two results contradict each other.

*No, events which are not connected to the model bow shock are not assigned any θBn at all, only the ones that are connected to the bow shock, which as explained above can have θBn>45°. There is no contradiction. The fact that SLAMS are found not to be connected to the bow shock may depend on several things; 1) The bow shock model has uncertainties, 2) there may be temporal variations of the bow shock position, 3) the large gyro radius may mean that parts of a SLAMS may not connect to the bow shock. We will add a brief discussion on this.*

d) One could argue that those found in the quasi-perpendicular region where not originated there but rather in the quasiparallel
region and later traveled to the other region. Actually, the authors make this point for the "isolated" SLAMS category, which sounds reasonable and could also explain -at least in part- why in Figure 9 the θBn is so broad.

*Yes, we agree that this is a possibility, similar to how we suggest that the isolated SLAMS are found outside of the general wave field.*

e) Line 244-249: the middle panel of Figure 11 includes only the connected events, while the other panels include´connected and not connected which prevents the reader from a direct comparison. The authors should show all panels for only the connected events. See my comment on lines 256-257.

*We do not agree, we think that it is interesting to see the distance also for 'non-connected' events. In practice the difference is very small, since the number of 'connected' events strongly dominate.*

f) Line 250: This result again sustains the fact that the probability of SLAMS in the foreshock region (connected to the shock, Table 2) is higher, but opposite to what is shown in Figure 7. This review has no other but to ask again for a response to such a dichotomy.

*We assume that the Reviewer means Figure 9, but SLAMS with a negative X$_f$ are not connected to the bow shock and, as we have discussed above, are not assigned a θ$_{Bn}$, and there is no contradiction with Figure 9.*

Line 256-257: The authors note the SLAMS they studied -similar to Earth- "are found on field lines connecting to the bow shock". Therefore SLAMS are foreshock structures and so they are

363 of such SLAMS which should have θBn angles typical for the quasi-parallel region upstream of the shock. The problem explained in my comments on Sec. 3.2.5 prevails here.

*Again, field lines connected to the bow shock do not necessarily have θBn<45°!*

Lines 267-274: What are the θBn angles of the 6 isolated events? Are they in the "connected to the shock" group? This reviewer thinks the connectivity to the shock would be more important to check before the type of structures, as it will be warranted that the structures are in the foreshock. Could the authors check for the 363 events connected to the bow shock how many structures are for each category? Is the isolated category still there?

Answers to questions like these can easily be found in the Supporting Information dataset. There it can be seen that five of six events are connected to the model bow shock, with $\theta_{Bn}$ between 17 and 76° (average of 48°). We think it is good scientific practice to first find the events that fulfill our definition of SLAMS, and not let exclude events based on our preconceived ideas of their properties. We rather view the connection nor otherwise to be a result of the study, and we find the most important aspect to be that a large majority of SLAMS are 'connected'. To answer the question on how many of each type are 'connected', see the table below (we have counted events inside the model bow shock as 'connected').

| Type | 'Connected' | 'Un-connected' |
|------|-------------|----------------|
| W | 314 | 37 |
| B | 24 | 2 |
| P | 18 | 1 |
| I | 5 | 1 |
| S | 15 | 2 |
| H | 37 | 3 |

Lines 284-286: How this ratio will give information on the wave growth rate? Please explain.

*This ratio says something about how long time the ULF waves will take to reach a non-linear stage after they enter the foreshock region, however we agree that perhaps this should rather be related to the ion foreshock boundary. We will add this caveat to the revised manuscript.*

Lines 287-295: One can argue that because of the weak bow shock at Mercury, in terms of its MA, the reflection of particles is less effective than at Earth. This would mean that waves possibly need more time to develop from the resonance of incident and backstreaming particles. The number of wave periods for a "geometrically equivalent" distance, and hence growth rate, would be lower at Mercury than at Earth. On the other hand, authors could find some references to the typical ULF waves frequency at Earth and Mercury, calculate the ratio and on the other hand calculate the number of wave periods separately for each planet. This way they could avoid "rough estimates" and have more realistic numbers.

*We agree with the Mach number/reflected ions argument, and will add this to the revised manuscript.*
*The spread around the typical values given by Eq 5 is large, and we feel that this estimate is a relevant first comparison. We will return to a more detailed investigation of the growth rates at a later study.*

Lines 297-301: In the cited works, indeed it is shown that some characteristics typical of a foreshock region can be found for angles θBn larger than 45°; however it is very clear that the

distribution of parameters peaks for 15-40° (see Glass et al., 2023). Events beyond θBn=50° are scarce and do not representative of the whole sample. This scenario differs widely from what is presented here in Figure 9 and cannot be compared. What is more, Blanco-Cano et al., 2009 reported ULF waves far from the nose of the shock for a radial IMF configuration and for low cone angle, θBV, that is still well within the foreshock; but do not report foreshock-like parameters/particles/waves for large θBn. In brief, the explanation for the unexpected behavior of θBn in Figure 9 is reasonably arguable.

*We agree that the results of Blanc-Cano et al. [2009] were perhaps over-interpreted by us, we will remove this reference. Instead, we refer to the paper by Le and Russell [1992], who report that the ULF wave boundary can extend to θBn = 60° for low cone angles at Earth. The situation may be similar at Mercury (although this remains to be studied.) Beyond these angles, we speculate that a gyro radius effect can be responsible. We will modify the discussion here.*

Lines 301-304: The authors could estimate the size of the SLAMS along the spacecraft trajectory, either using a proxy (case by case) or an average of the Vsw. Normalized to Rm would give a better idea of how extended are the structures in order to help sustain this alternative explanation. Actually, one can take the 1.2s with an average VSW~460 km/s (Diego et al. 2020 and references therein), we found the extension to be 0.22RM. This reviewer finds it hard to think that a SLAMS with a cross-section of (at least) 0.02 RM located in the quasi-parallel region could be extended into the quasi-perpendicular shock. At most, one could think that if this SLAM is located at the edge of the foreshock then the statement is true; but according to the statistical results in Figure 9, there would be a very large number of SLAMS with this configuration.

*The scale size calculation by the Reviewer refers to the extent parallel to the solar wind velocity. It is likely that the scale size in other directions is considerably larger, if SLAMS retain the properties of ULF waves at least to some extent (see the Figure below from simulations by Jarvinen et al., 2019). (We are at the moment investigating this at Earth, using Cluster multipoint measurements.) Together with finite gyro radius effects associated with the heated reflected ions, this may be a possible scenario.*

[Figure]

*From Jarvinen et al. [2019].*

Lines 305-313: In lines 286-293 and after some rough estimations on the wave growth rate in Mercury and Earth foreshocks, the authors conclude that "the non-linear development of SLAMS should be comparable between Earth and Mercury". Such result go against the new interpretation presented now in the discussion section.

*We do not think that there is an inconsistency. The necessary conditions for SLAMS generation may be more rare at Mercury, but once they exist, the growth rates may be similar. Another way to think about this can be that the growth rate is generally slower on Mercury, but that the high-rate tail of a distribution can give rise to SLAMS at Mercury, but then more rarely than at Earth. These events are the ones we observe with MESSENGER, and we therefore conclude that the growth rates are similar to those at Earth. We will change the discussion slightly along these lines.*

Line 338-339: As mentioned before this conclusion is reasonably arguable and should be adapted after a detailed revision of all the reviewer comments.

*We do not see any reason to change the conclusions in these lines. They are arguably typically found within a region of general ULF wave activity (these are the 'W' class events), and a large majority of the events connect magnetically to a model bow shock. We can argue about the reasons, but these are observational facts.*

**Minor comments**

We agree with the minor comments and will take them into account in the future revision.

**References**

Diego, P., Piersanti, M., Laurenza, M., & Villante, U. (2020). Properties of solar wind structures at Mercury's orbit. Journal of Geophysical Research: Space Physics, 125(9), e2020JA028281.

Eastwood, J. P., Lucek, E. A., Mazelle, C., Meziane, K., Narita, Y., Pickett, J., & Treumann, R. A. (2005). The foreshock. Space Science Reviews, 118, 41-94.

Hanneson, C., Johnson, C. L., Mittelholz, A. M., Al Asad, M. M., & Goldblatt, C. (2020). Dependence of the interplanetary magnetic field on heliocentric distance at 0.3–1.7 AU: A six-spacecraft study. Journal of Geophysical Research: Space Physics, 125, e2019JA027139. https://doi.org/ 10.1029/2019JA027139

Jarvinen, R., Alho, M., Kallio, E., & Pulkkinen, T. I. (2020). Ultra-low-frequency waves in the ion foreshock of Mercury: a global hybrid modelling study. Monthly Notices of the Royal Astronomical Society, 491(3), 4147-4161.

Kis, A., Scholer, M., Klecker, B., Kucharek, H., Lucek, E. A., & Rème, H. (2007, March). Scattering of field-aligned beam ions upstream of Earth's bow shock. In Annales Geophysicae (Vol. 25, No. 3, pp. 785-799). Copernicus GmbH.

Le, G., & Russell, C. T. (1992). A study of ULF wave foreshock morphology—I: ULF foreshock boundary. Planetary and space science, 40(9), 1203-1213.

Le, G., Chi, P. J., Blanco-Cano, X., Boardsen, S., Slavin, J. A., Anderson, B. J., & Korth, H. (2013). Upstream ultra-low frequency waves in Mercury's foreshock region: MESSENGER magnetic field observations. Journal of Geophysical Research: Space Physics, 118(6), 2809-2823.

Romanelli, N., DiBraccio, G., Gershman, D., Le, G., Mazelle, C., Meziane, K., ... & Espley, J. (2020). Upstream ultra-low frequency waves observed by MESSENGER's magnetometer: Implications for particle acceleration at Mercury's bow shock. Geophysical Research Letters, 47(9), e2020GL087350.

Romanelli, N., & DiBraccio, G. A. (2021). Occurrence rate of ultra-low frequency waves in the foreshock of Mercury increases with heliocentric distance. Nature Communications, 12(1), 6748.

Schwartz, S. J., Burgess, D., Wilkinson, W. P., Kessel, R. L., Dunlop, M., & Lühr, H. (1992). Observations of short large-amplitude magnetic structures at a quasi-parallel shock. Journal of Geophysical Research: Space Physics, 97(A4), 4209-4227.

Sundberg, T., Boardsen, S. A., Slavin, J. A., Uritsky, V. M., Anderson, B. J., Korth, H., ... & Solomon, S. C. (2013). Cyclic reformation of a quasi-parallel bow shock at Mercury: MESSENGER observations. Journal of Geophysical Research: Space Physics, 118(10), 6457-6464.

Szabo, A., Larson, D., Whittlesey, P., Stevens, M. L., Lavraud, B., Phan, T., ... & Pulupa, M. (2020). The heliospheric current sheet in the inner heliosphere observed by the parker solar probe. The Astrophysical Journal Supplement Series, 246(2), 47.

Tsurutani, B. T., Lakhina, G. S., Pickett, J. S., Guarnieri, F. L., Lin, N., & Goldstein, B. E. (2005). Nonlinear Alfvén waves, discontinuities, proton perpendicular acceleration, and magnetic holes/decreases in interplanetary space and the magnetosphere: intermediate shocks?. Nonlinear Processes in Geophysics, 12(3), 321-336.

Uritsky, V. M., Slavin, J. A., Boardsen, S. A., Sundberg, T., Raines, J. M., Gershman, D. J., ... & Korth, H. (2014). Active current sheets and candidate hot flow anomalies upstream of Mercury's bow shock. Journal of Geophysical Research: Space Physics, 119(2), 853-876.